# Revisiting Discover-then-Name Concept Bottleneck Models: A Reproducibility Study

**Freek Byrman**                                                                      *freek.byrman@student.uva.nl*
*University of Amsterdam*

**Emma Kasteleyn**                                                                *emma.kasteleyn@student.uva.nl*
*University of Amsterdam*

**Bart Kuipers**                                                                      *bart.kuipers@student.uva.nl*
*University of Amsterdam*

**Daniel Uyterlinde**                                                            *daniel.uijterlinde@student.uva.nl*
*University of Amsterdam*

**Reviewed on OpenReview:** *https:// openreview. net/ forum? id=946cT3Jsq5*

## Abstract

Concept Bottleneck Models (CBMs) (Koh et al., 2020) are a class of interpretable deep learning frameworks that improve transparency by mapping input data into human-understandable concepts. Recent advances, including the Discover-then-Name CBM proposed by Rao et al. (2024), eliminate reliance on external language models by automating concept discovery and naming using a CLIP feature extractor and sparse autoencoder. This study focuses on replicating the key findings reported by Rao et al. (2024). We conclude that the core conceptual ideas are reproducible, but not to the extent presented in the original work. Many representations of active neurons appear to be misaligned with their assigned concepts, indicating a lack of faithfulness of the DN-CBM's explanations. To address this, we propose a model extension: an enhanced alignment method that we evaluate through a user study. Our fine-tuned model provides more interpretable concepts (with statistical significance), at the cost of a slight decrease in accuracy. The implementation of our work is available in our [GitHub repository](#).

## 1 Introduction

Interpretable frameworks like Concept Bottleneck Models (CBMs) (Koh et al., 2020) have gained attention for their ability to enhance explainability in deep learning. CBMs accomplish this by mapping input data into a human-understandable concept space, which can then be used for downstream tasks, such as classification. This is achieved through a linear combination of concepts, which allows for explaining the predictions made by a classifier. Traditional CBMs require labeled attribute datasets, but recent CBMs (Oikarinen et al., 2023; Yang et al., 2023) use Large Language Models (LLMs) (Brown et al., 2020) and vision-language models (VLMs) (Radford et al., 2021) for attribute-label-free concept learning. While effective, these methods have several limitations: They require creating concept sets for each new classification task, depend on prompt engineering, are inherently stochastic, and may fail to produce key concepts needed for accurate classification.

To address these limitations, Rao et al. (2024) propose the Discover-then-Name CBM (DN-CBM), which automates the discovery and naming of concepts without relying on external LLMs. It uses a CLIP-based feature extractor (Radford et al., 2021) and Sparse Autoencoder (SAE) to disentangle input embeddings into human-understandable concepts for classification. Rao et al. (2024) report that the DN-CBM model not only provides interpretability but also achieves competitive performance in terms of classification accuracy. Notably, it outperforms state-of-the-art explainable baseline models, such as Yang et al. (2023), across most

datasets, while remaining task-agnostic. This compelling result motivates our reproduction to further explore its mechanisms and contribute to advancements in explainable artificial intelligence.

In this study, we reproduce and evaluate the findings presented by Rao et al. (2024), focusing on the performance and explainability of the DN-CBM framework. Building on their findings, we investigate the impact of cosine similarity on concept explainability and introduce a loss function that promotes more interpretable concepts. This loss function encourages alignment between the SAE neurons and their assigned concepts, which we then evaluate through a user study.

## 2 Scope of reproducibility

We investigate the following (main) claims from Rao et al. (2024) and label them (**C1**-**C3**) for reference.

- **C1**: **Automated concept discovery**. The DN-CBM framework can successfully discover latent concepts in the data without pre-selecting them. The SAE effectively identifies meaningful and human-understandable concepts directly from the CLIP feature space.

- **C2**: **Interpretability**. The method demonstrates that the discovered dictionary vectors align well with text embeddings of the concepts they represent in CLIP space. This alignment enables intuitive naming of the concepts, facilitating model interpretability across different tasks. As a result, the approach supports task-agnostic explanations of the model's decision process.

- **C3**: **Performance**. The DN-CBM achieves competitive performance on classification tasks across a variety of downstream datasets, ensuring task-agnosticity.

## 3 Methodology

This section outlines the methods used in this study. Sections 3.1, 3.2, and 3.3 discuss the models, datasets, and hyperparameter considerations, respectively. Section 3.4 details the experimental setup used to validate the claims, and Section 3.5 presents the motivation, theoretical foundation, and evaluation of our extension.

### 3.1 Model descriptions

**Concept discovery**. We follow the SAE approach proposed by Bricken et al. (2023) to transform CLIP features into a more interpretable latent space. This is achieved using a linear encoder $f(\cdot)$ with weights $\boldsymbol{W}_E \in \mathbb{R}^{d \times h}$, followed by a ReLU activation function $\phi(\cdot)$. The SAE is trained in a self-supervised manner by reconstructing the original features using a linear decoder $g(\cdot)$ with weights $\boldsymbol{W}_D \in \mathbb{R}^{h \times d}$. The latent space dimension, denoted by $h$, is much larger than the CLIP embedding dimension, denoted by $d$. For a given embedding $\boldsymbol{a} \in \mathbb{R}^d$, that is produced by a CLIP image encoder $\mathcal{I}$, the loss function is defined as:

$$\mathcal{L}_{\text{SAE}}(\boldsymbol{a}) = \|\text{SAE}(\boldsymbol{a}) - \boldsymbol{a}\|_2^2 + \lambda_1 \|\phi(f(\boldsymbol{a}))\|_1, \tag{1}$$

where $\lambda_1$ is a hyperparameter that enforces sparsity in activations. The SAE is typically trained on a large dataset, denoted as $\mathcal{D}_{\text{extract}}$, to extract a wide range of concepts.

**Concept naming**. After training, we automatically assign names to individual feature dimensions in the SAE's hidden representation. To achieve this, a vocabulary set $\mathcal{V} = \{v_1, \ldots, v_{|\mathcal{V}|}\}$ is embedded using a CLIP text encoder $\mathcal{T}$. To enable effective generalization and meaningful concept naming, the set $\mathcal{V}$ should be broad and flexible, for example, a large collection of unigrams. Neurons in the SAE's latent space are labeled according to the highest cosine similarity between their dictionary (decoder) weights and the CLIP vocabulary representations. This is a natural choice as CLIP was trained to optimize cosine similarities between text and image embeddings. The dictionary weight vector $\boldsymbol{p}_c$ for neuron $c$ is defined as the $c^{th}$ row of the decoder weight matrix:

$$\boldsymbol{p}_c = [\boldsymbol{W}_D]_{c,:} \in \mathbb{R}^d. \tag{2}$$

The corresponding label $s_c$ is then determined as:

$$s_c = \underset{v \in \mathcal{V}}{\operatorname{argmax}} \cos\left(\angle\left(\boldsymbol{p}_c, \mathcal{T}(v)\right)\right). \tag{3}$$

We define a vector representing alignment as a distribution over all cosine similarities so that we can refer to this later. Specifically, we introduce a vector $\boldsymbol{v} \in \mathbb{R}^h$, where the $c^{th}$ element is defined as:

$$v_c = \cos\left(\angle\left(\boldsymbol{p}_c, \mathcal{T}(s_c)\right)\right). \tag{4}$$

We refer to $v_c$ as the *cosine similarity score* of concept $c$ throughout this work. It quantifies the alignment between the assigned vector in CLIP space and the dictionary vector of neuron $c$.

**Constructing CBMs.** A CBM is constructed by connecting a linear probe $h(\cdot)$ to the encoder's output for downstream classification tasks. The probe is trained on a separate dataset, denoted $\mathcal{D}_{probe} = \{(\boldsymbol{x_1}, y_1)\}, (\boldsymbol{x}_2, y_2), \dots \}$, where $y_i$ represents the ground truth label of $\boldsymbol{x}_i$. The CBM $t(\cdot)$ is defined as:

$$t(\boldsymbol{x}_i) = (h \circ \phi \circ f \circ \mathcal{I})(\boldsymbol{x}_i). \tag{5}$$

During probe training, the SAE encoder layer is frozen, and the probe weights ($\boldsymbol{\omega}$) are adjusted based on the following loss function, where $\lambda_2$ is a sparsity hyperparameter and CE denotes the cross-entropy loss:

$$\mathcal{L}_{\text{probe}}(\boldsymbol{x}_i) = \text{CE}\left(t(\boldsymbol{x}_i), y_i\right) + \lambda_2 |\boldsymbol{\omega}|_1. \tag{6}$$

## 3.2 Datasets

CC3M is used for training and evaluating the SAE ($\mathcal{D}_{\text{extract}}$), whereas ImageNet, Places365*, CIFAR10, CIFAR100, and Waterbirds-100 are used for training and evaluating the linear probe ($\mathcal{D}_{\text{probe}}$):

**CC3M**. CC3M consists of image-caption pairs generated by extracting text from alt-texts of images on the web (Sharma et al., 2018). Due to link rot, approximately 68% of the originally collected images were available in the final dataset. The dataset can be downloaded here.

**ImageNet**. ImageNet-1K is a standard benchmark for image classification (Deng et al., 2009).

**Places365**. We use a 10% subset of the Places365 dataset (Zhou et al., 2017), sampled to preserve the original class distribution, for downstream classification. We refer to this subset as Places365*.

**CIFAR100**. CIFAR100 contains 60,000 images evenly distributed across 100 classes (Krizhevsky, 2009).

**Waterbirds-100**. The Waterbirds-100 dataset (Petryk et al., 2022; Sagawa et al., 2020) features landbirds and waterbirds with spurious background correlations during training, but not in the test set.

## 3.3 Hyperparameters

We reproduce the experiments using the hyperparameters from Rao et al. (2024), as reported in Table 1, with $v_2$ as the default for standard classification and $v_3$ for concept interventions, unless otherwise specified.

**Table 1: Hyperparameters**. We use the same hyperparameters as Rao et al. (2024), though the original paper describes a hyperparameter sweep without detailing the probe settings. For standard classification, we evaluate two linear probe configurations: $v_1$, based on the GitHub README example for Places365, and $v_2$, the default settings provided in the code for each probe dataset. A third configuration, $v_3$, is used specifically for concept intervention experiments. The Adam optimizer is applied with its default hyperparameter settings (Kingma and Ba, 2015).

| General | | SAE | | Probe | | | |
|---|---|---|---|---|---|---|---|
| Hyperparameter | Value | Hyperparameter | Value | Hyperparameter | $v_1$ | $v_2$ | $v_3$ |
| text encoder ($\mathcal{T}$) | CLIP ResNet-50 | latent dim ($h$) | 8192 | learning rate[1] | $10^{-2}$ | $10^{-3}$ | $10^{-1}$ |
| image encoder ($\mathcal{I}$) | ResNet-50 | $L_1$ sparsity ($\lambda_1$) | $3 \times 10^{-5}$ | batch size | 512 | 512 | 512 |
| vocabulary ($\mathcal{V}$) | Google 20k | learning rate | 0.1 | epochs | 200 | 200 | 200 |
| vocabulary size ($|\mathcal{V}|$) | 20000 | epochs | 200 | $L_1$ sparsity ($\lambda_2$) | 0.1 | 1 | 10 |
| embedding dim ($d$) | 1024 | batch size | 4096 | optimizer | Adam | Adam | Adam |
| | | batch resample freq | 10 | top-$k$ pruning | - | - | 5 |
| | | optimizer | Adam | | | | |

[1]For CIFAR100, a probe learning rate of $10^{-2}$ was used in $v_2$.

### 3.4 Experimental setup and code for reproducibility

We used the publicly available codebase by Rao et al. (2024) to reproduce the key findings of the original study. This comprehensive [GitHub repository](#) includes all necessary scripts to generate the plots and the final results. The computational tasks were carried out using GPU resources provided by the Dutch national supercomputer, Snellius, with funding support from the University of Amsterdam. An Nvidia A100 Tensor Core GPU was used to run the experiments. The total computational expense for the reproduction equals 91.91 GPU hours. This has an estimated emissions of 10.87 kgCO2eq.[2]

To investigate **C1**, we train an SAE with the same architecture and hyperparameters as the original paper. We then align the dictionary vectors (Equation 2) with the CLIP feature representations. We visualize the ranked cosine similarity scores for the DN-CBM from a checkpoint in the GitHub repository and our reproduced DN-CBM. This will help us determine if our reproduced latent space is as closely associated with CLIP features as in the original work. We use this same figure as an initial indication for **C2**, which asserts that the vectors should align well. To further investigate **C2**, we reproduce qualitative and quantitative analyses from Rao et al. (2024) related to interpretability.

The qualitative analyses that we reproduce for **C2** comprise illustrating examples of named concepts alongside the top images that activate these concepts across four datasets, as well as explaining the decision of the DN-CBM by classifying random images from the Places365* dataset and reporting the top contributing concepts. While Rao et al. (2024) include only concepts with high cosine similarity scores in their analysis, we extend this by also incorporating lower-aligned concepts to assess their representational quality. We further evaluate generalizability by providing similar results for additional datasets in Appendix B.1.

We reproduce the two components of the quantitative analyses for **C2**. First, the human feedback survey is reproduced to validate the alignment between concepts and neurons across varying cosine similarity scores. Participants were asked to rate concept consistency and naming accuracy for 12 images under one concept. Similar to the qualitative analysis, these are the top images that activate the concept. High consistency is defined as a set of images with a consistent overarching theme. The survey employs a 1-5 rating scale for accuracy, where 1 indicates poor alignment and 5 indicates strong alignment between the concept and the images. This survey was conducted with 22 participants, following the exact structure of the original study by Rao et al. (2024).

Second, concept interventions are evaluated using the Waterbirds-100 dataset, which is specifically designed to test robustness to spurious correlations between bird type and background, evaluating **C2**. In this dataset, landbirds (waterbirds) are shown on land (water) backgrounds during training, while such correlations are absent in the test set. We replicate this setup by training on Waterbirds-100 and applying two interventions to the DN-CBM model: (1) retaining only bird-related concepts, and (2) removing them. We assess classification accuracy before and after these interventions, focusing on their impact on 'worst-group' examples: Landbirds on water- and waterbirds on land backgrounds.

To reproduce **C3**, we follow the original DN-CBM model specification by attaching a linear probe to the SAE to classify images. We compute the classification accuracy on ImageNet, Places365*, CIFAR10, and CIFAR100, specified in Section 3.2. We compare our accuracies to the original accuracies.

We omit certain additional analyses from Rao et al. (2024), as they are not essential for verifying the main claims. One such analysis is a quantitative evaluation using the SUNAttributes dataset (Patterson et al., 2014), where discovered concepts from the SAE are compared to ground truth labels by filtering and merging nodes based on cosine similarity with text embeddings. We did not reproduce this evaluation, as it constitutes a minor aspect of the original work and employs a filtering approach not used elsewhere in the study. We also omit the semantic consistency analysis using $k$-means clustering on concept activation vectors and the CLIP-Dissect component of the user study. These analyses, while informative, are not critical for assessing the main contributions.

---

[2]The emissions were calculated using [Machine Learning calculator](#) (Lacoste et al., 2019). The estimated carbon efficiency of the Netherlands was 0.473 kgCO2eq in January 2025 (Nowtricity, 2025).

### 3.5 Extension to original work

In their novel method, Rao et al. (2024) propose a post-hoc approach to explainable artificial intelligence. This approach is compelling for maintaining the model's accuracy, as it does not constrain the model to be inherently explainable during training. However, the degree of explainability is theoretically limited as certain aspects of a freely trained model cannot be fully captured by a single word or concept due to the constraints of human language. Rao et al. (2024) report good alignment between dictionary vectors and text embeddings in CLIP space (high cosine similarity score). However, defining "good" alignment in high-dimensional space is challenging, especially when the alignment varies significantly across the neurons. We investigate this alignment in **C2** and propose a model extension to improve the alignment of dictionary vectors and text embeddings.

#### 3.5.1 Model extension

We introduce a loss function that encourages SAE neurons to align with more interpretable concepts. This loss function is designed for fine-tuning, as it depends on a set of pre-defined concepts learned during training. This loss function is defined as:

$$\mathcal{L}_{\text{FSAE}}(\boldsymbol{a}) = \|\text{SAE}(\boldsymbol{a}) - \boldsymbol{a}\|_2^2 + \lambda_1 \|\phi(f(\boldsymbol{a}))\|_1 - C\left(\frac{\phi(f(\boldsymbol{a}))}{\max(\|\phi(f(\boldsymbol{a}))\|_2, \epsilon)} \cdot \boldsymbol{v}\right), \tag{7}$$

where $C$ is the cosine penalty parameter that dictates the strength of our introduced penalty term, and $\epsilon$ is a small positive constant added to prevent division by zero in cases with no activations. The term $C$ encourages alignment between the latent space and explainable features using the cosine similarity scores of $\boldsymbol{v}$. The conceptual idea behind our fine-tuning loss is shown in Figure 1.

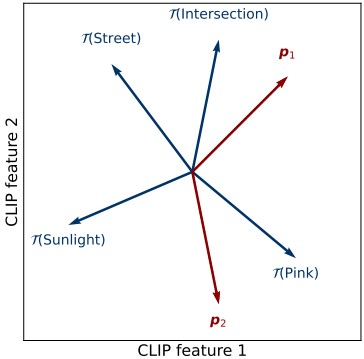
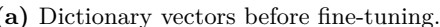

(a) Dictionary vectors before fine-tuning.

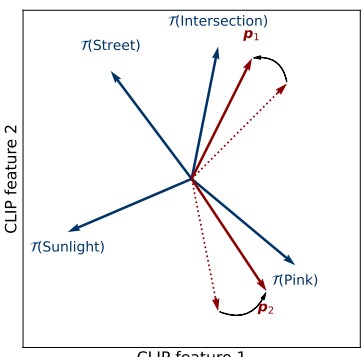

(b) Dictionary vectors after fine-tuning.

**Figure 1: Conceptual overview of the fine-tuning process, in CLIP embedding space.** In this example, the vocabulary set is $\mathcal{V} = \{\text{Street}, \text{Pink}, \text{Intersection}, \text{Sunlight}\}$, with both CLIP and latent dimensions $h = d = 2$. Neuron 1 is assigned $s_1 = \{\text{Intersection}\}$, and neuron 2 is assigned $s_2 = \{\text{Pink}\}$. After fine-tuning, the dictionary vectors are better aligned with the CLIP embeddings of their respective assigned names. In practice, the vocabulary set, CLIP embedding dimension, and number of neurons are much larger.

A key advantage of scaling by $\phi(f(\boldsymbol{a}))$ is that the incentive for alignment is proportional to the activation magnitude. This prevents rarely activated neurons from arbitrarily aligning with CLIP embeddings without capturing meaningful representations. Consequently, the neurons involved in inference are more likely to exhibit a high cosine score. The normalization prevents the optimization routine from generating excessively large activations to exploit the cosine loss.

Excessively large $C$ values, however, introduce an issue. When $C$ becomes too large, the cosine similarity term dominates the loss function, causing the distribution of $\boldsymbol{v}$ to shift towards higher values. This reduces variation in similarity values. As more explainable neurons are incentivized to activate, their activations

may lose meaning. Furthermore, because we normalize $C\left(\phi(f(\boldsymbol{a})) \cdot \boldsymbol{v}\right)$ by $\|\phi\left(f(\boldsymbol{a})\right)\|$, there is no incentive to increase the activation magnitude. This leads to the diffusion of neuron activations, meaning that more neurons become active but with lower activation values. This reduces the model's interpretability because when it relies on a vast number of weakly activated neurons, it becomes difficult to pinpoint specific neurons as the primary contributors to the decision-making process. This creates a trade-off: increasing neuron explainability can sometimes come at the cost of making their activations less meaningful.

To determine an appropriate value for $C$, we experiment with different values and analyze their impact on neuron interpretability, activations, and accuracy. We fine-tune our model seven times on CC3M, each time using a different value of the hyperparameter $C \in [10^{-6}, 10^{-5}, \ldots, 10^{0}]$. We use the mean cosine similarity score as a proxy for neuron interpretability, while activation levels are quantified by the average magnitude of nonzero elements in $\phi(f(\boldsymbol{a}))$. Additionally, we measure the validation accuracy on the Places365* dataset. A balanced value of $C$ is then selected based on a qualitative assessment of these factors. The code for our reproduction and extension is publicly available in our [GitHub repository](#).

### 3.5.2 Evaluation model extension

To evaluate the impact of our method, we compared the reproduced DN-CBM model with our fine-tuned extension using the optimal $C$ value. We do this by conducting a user study. For our user study, we selected all concept explanations for image classifications that appeared more than five times in a subset of the Places365* test set. Next, we ranked these concepts by cosine similarity and selected the bottom 40, middle 40, and top 40 aligning concepts for both models (resulting in a total of 6 groups). For each concept, we sampled five images without replacement corresponding to the classification explanation, resulting in a total of 240 concepts, each paired with five images. We asked each participant to rate two randomly sampled concepts from each group on a scale from 0 to 5, reflecting how many images aligned with the concept. This sampling approach ensured that each participant evaluated only 12 images while the entire dataset was thoroughly assessed across all participants through the randomized selection process. A total of 203 participants completed the questionnaire. Further details of the outline of our user study are given in Appendix A.1. To assess whether the differences in user ratings between the two models are statistically significant, we employ the Wilcoxon signed-rank test (Wilcoxon, 1945). This non-parametric test is appropriate given the paired nature of our data and the lack of a normal distribution in the ratings. The details of the test and its implementation are further discussed in Appendix A.2.

Note that this user study, conducted to evaluate our model extension, is the second user study in this work and differs from the first, which followed the exact structure of the study by Rao et al. (2024). In their work, they analyzed neuron activations in response to concepts. We argue that classification explanations are more meaningful for our objective because they align with the model's primary task. For example, a neuron representing the concept of "turquoise" may have a list of top-activating images, such as a car in CIFAR10. However, this does not necessarily indicate that the neuron plays a key role in classifying the object (car).

We also include classification experiments on CIFAR10, CIFAR100, Places365*, and ImageNet, and assess concept intervention effectiveness on Waterbirds-100 to evaluate our extension.

## 4 Results

This section begins by presenting findings on the reproducibility of **C1** through **C3**. Subsequently, the results of extensions to DN-CBM are discussed. The findings confirm **C1** and **C3**, while contradictory results are observed for **C2**.

### 4.1 Results reproducing original paper

To assess the reproducibility of **C1** we visualize the ranked cosine scores for the original and reproduced DN-CBM in Figure 2. We observe that we successfully reproduce a latent space that maps concepts within a certain cosine similarity range. The distribution of values closely aligns with the original findings, demonstrating that our approach achieves a comparable mapping. This validates the reproducibility of **C1**.

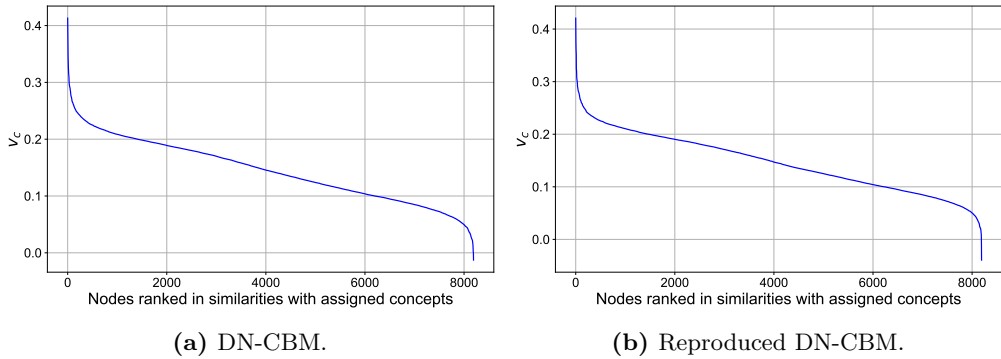

**(a)** DN-CBM.            **(b)** Reproduced DN-CBM.

**Figure 2: Ranked cosine similarity scores of the assigned concepts**. Comparison of the cosine similarity scores from the original DN-CBM (Figure 2a) and the reproduced DN-CBM (Figure 2b)

To validate **C2**, we extracted the top-activating images with the highest cosine similarity score, across the four datasets, as shown in Figure 3. Notably, the top-activating images strongly correspond to their respective concepts for ImageNet and Places365, indicating a high degree of semantic consistency. This is in line with the results of Rao et al. (2024) and supports **C2**. For CIFAR10 and CIFAR100, the images correspond less with the concepts "plaid" and "sweater", which may be attributed to the limited expressiveness of the dataset.

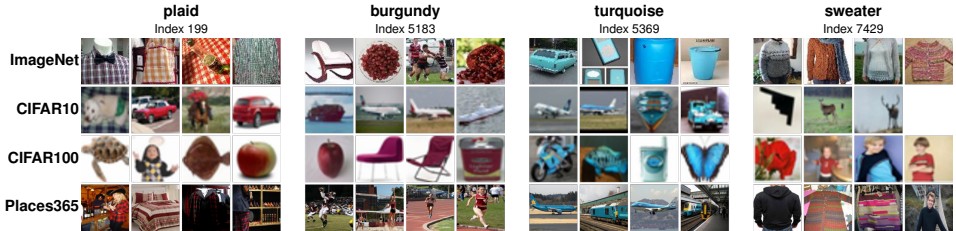

**Figure 3: Task-agnosticity of concept extraction**. We present the concepts with the highest cosine similarity score, alongside their top-activating images from four datasets.

To further assess the reproducibility of **C2**, we examine Figure 2. The cosine scores range broadly from approximately $-0.01$ to $0.42$, indicating the presence of lower-aligned concepts. This raises concerns on the faithfulness of the interpretability of some notes. The original study focuses only on highly aligned concepts (Figure 3), which may not generalize to the explainability of all neurons. To assess **C2** across the entire network, we examine Figure 4, which presents concepts with lower cosine similarity scores and their top-activating images across four datasets. These images demonstrate low consistency with the assigned concept names, which suggests that **C2** is not fully supported by this figure, as the images should only activate for concepts they actually represent.

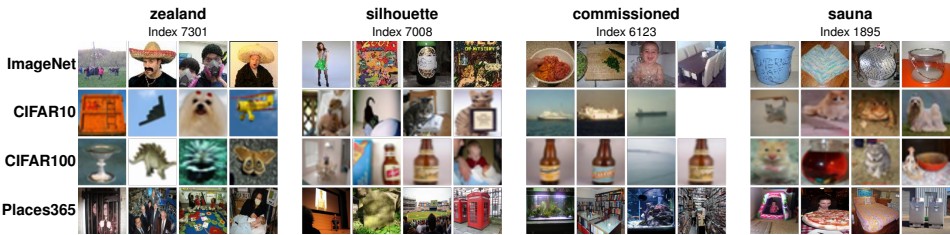

**Figure 4: Lower aligned task-agnosticity of concept extraction**. We present low-aligned concepts alongside their top-activating images from four datasets. The images associated with each concept demonstrate low consistency with the assigned concept name across datasets.

To continue the qualitative analysis of **C2**, we display the top concepts contributing to the decision-making process for two randomly selected samples from the Places365* dataset in Figure 5. Rao et al. (2024) show similar examples with concepts that, indeed, all describe aspects of the image's theme. This qualitative analysis supports their claim that concepts are associated with the predicted class, thus aiding interpretability. Upon examining Figure 5, we find that not all displayed concepts contribute meaningfully to the decision-making process, which partially challenges **C2**. The left figure illustrates that the concepts are thematically consistent with the corresponding class, supporting the validity of the approach. The right figure raises concerns. Specifically, the presence of concepts such as "kayaking", "dams", and "trivium", do not clearly correspond to the class. Consequently, we find that the original results cannot be reproduced to the same extent. This qualitative analysis is extended to different datasets in Appendix B.1, which demonstrates similar results. One final observation related to Figure 5 is that a local explanation, such as "greenhouse," directly mirrors the class label "greenhouse". Explaining a prediction with its own class label, however, offers limited insight; this highlights a general limitation of the DN-CBM framework.

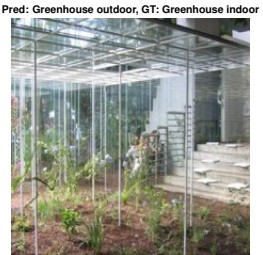 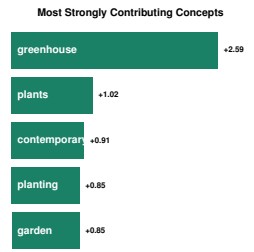 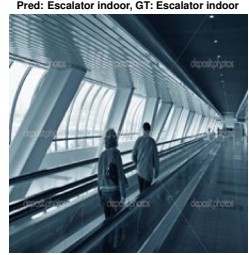 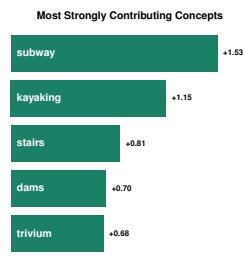

**Figure 5: Explaining decisions using the reproduced DN-CBM**. We present randomly drawn examples of images from the Places365* dataset alongside the top concepts contributing to their classification.

To reproduce the quantitative analyses for **C2**, the survey and concept interventions results are presented in Figure 6 and Table 2 respectively. Figure 6a shows a general decline in semantic consistency with decreasing concept alignment. The intermediate alignment trend differs slightly from the original work, likely due to limited sample sizes. Figure 6b reveals a positive relation between accuracy and consistency, though some low-accuracy cases still exhibit high consistency. Overall, the trends align with the original results, indicating reasonable reproducibility. The concept interventions in Table 2 demonstrate effectiveness, consistent with the original findings.

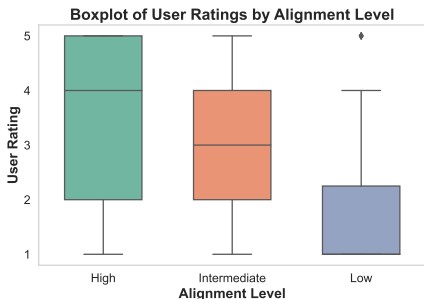 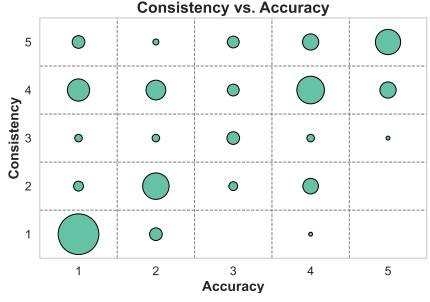

**(a)** Semantic consistency of concepts.    **(b)** Consistency vs. accuracy.

**Figure 6: Reproduced user study on concept accuracy**. Semantic consistency is plotted for nodes with high, intermediate, and low alignment to their assigned text embeddings of the reproduced SAE (Figure 6a). In Figure 6b, the semantic consistency scores are plotted against name accuracy. The survey has 22 participants.

To assess **C3**, we report the classification accuracy for the reproduced model and the original work in Table 3. The classification accuracies of ImageNet, CIFAR10 and CIFAR100 are similar for the original and reproduced DN-CBM. The performance on Places365* is worse for the reproduced model. This could be attributed to the use of a smaller version of this dataset. Overall, our results are in agreement with **C3**.

**Table 2: Concept interventions**. Classification accuracy (%) is reported on the full test set ("Overall") and four specific groups before and after interventions. "Reproduced" uses configuration $v_3$, while "Fine-tuned" uses $v_3$ with $\lambda_2 = 1$ (vs. 10). Details and the intervened concepts are provided in Appendix C.1.

| Source | Model | Overall | Worst groups | | Training groups | |
|---|---|---|---|---|---|---|
| | | | L.Bird@W | W.Bird@L | L.Bird@L | W.Bird@W |
| Reproduced | Before intervention | 84.24 | 76.30 | 58.27 | 97.83 | 91.37 |
| | Only bird concepts | 88.07 (+3.8) | 84.78 (+8.5) | 82.73 (+24.5) | 93.28 (-4.6) | 87.05 (-4.3) |
| | Non-bird concepts | 59.47 (-24.8) | 28.70 (-47.6) | 15.83 (-42.4) | 97.40 (-0.4) | 79.14 (-12.2) |
| Fine-tuned | Before intervention | 67.39 | 42.39 | 19.42 | 99.57 | 91.37 |
| | Only bird concepts | 83.24 (+15.8) | 73.70 (+31.3) | 75.54 (+56.1) | 92.19 (-7.4) | 92.81 (+1.4) |
| | Non-bird concepts | 65.14 (-2.3) | 43.70 (+1.3) | 4.32 (-15.1) | 100.00 (+0.4) | 81.29 (-10.1) |

**Table 3: Comparison of performance of the original and reproduced results**. We report the classification accuracy (%) of the original paper and our reproduction using CLIP ResNet-50 on ImageNet, Places365*, CIFAR10 and CIFAR100. "Fine-tuned" is our model suggestion, which incorporates a cosine loss function with $C = 10^{-4}$ (Equation 7). '*' indicates the use of a smaller dataset compared to the original paper; 10% of Places365. Results show mean ± standard deviation over 3 probe training runs.

| Model | ImageNet | Places365 | CIFAR10 | CIFAR100 |
|---|---|---|---|---|
| DN-CBM (Rao et al.) | 72.9 | 53.5 | 87.6 | 67.5 |
| DN-CBM Reproduced | 72.65 ±0.02 | 49.96* ±0.04 | 86.71 ±0.02 | 68.51 ±0.04 |
| DN-CBM Fine-tuned (Ours) | 70.47 ±0.04 | 49.30* ±0.03 | 83.88 ±0.02 | 64.39 ±0.16 |

## 4.2 Results beyond original paper

Figure 7 shows the results of our experiments with different cosine penalty parameters, $C$. After completing standard training, we fine-tuned for an additional 30 epochs using the loss function in Equation 7, which was generally sufficient for convergence. We observe the theoretical trends discussed in Section 3.5. As $C$ increases, activations become more diffused, the average cosine similarity score rises, and at some point, it comes at the cost of accuracy. Figure 8 shows the cosine score distributions for $C = 10^{-3}$ and $C = 10^{-4}$, both of which are significantly higher on average compared to Figure 2. The more diffused activations result in a sharper distribution, as seen in Figure 8b. Appendix C.2 provides a visualization of the cosine score distribution across all tested $C$ values.

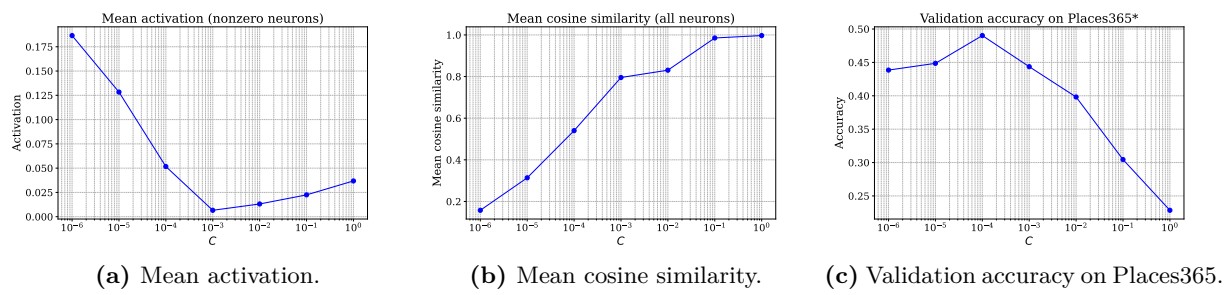

(a) Mean activation.   (b) Mean cosine similarity.   (c) Validation accuracy on Places365.

**Figure 7: Impact evaluation of $C$**. Comparison of mean activation across nonzero neurons, mean cosine score, and accuracy for different $C$ values. These plots are obtained with probe hyperparameters $v_1$.

Unfortunately, we lack a straightforward quantitative metric for determining the optimal value of $C$. We recommend using the theoretical considerations discussed earlier to guide its selection. For all remaining experiments, we use $C = 10^{-4}$, which offers a balanced trade-off, substantially improving the average cosine similarity (from 0.146 to 0.540), while maintaining reasonable activation patterns and achieving the highest validation accuracy. Our goal is to provide a proof of concept that demonstrates how promoting alignment

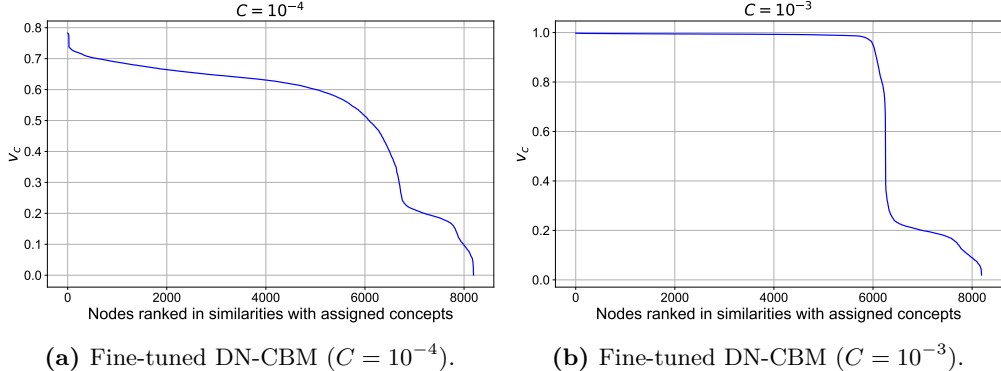

**(a)** Fine-tuned DN-CBM ($C = 10^{-4}$). **(b)** Fine-tuned DN-CBM ($C = 10^{-3}$).

**Figure 8: Ranked cosine similarity scores of the assigned concepts after fine-tuning**. Comparison of cosine score distributions after fine-tuning with penalties $C = 10^{-3}$ (Figure 8a) and $C = 10^{-4}$ (Figure 8b).

can lead to more faithful and interpretable model explanations. We encourage future work to develop more principled, quantitative approaches for selecting $C$.

Referring to Table 3, we compare the accuracy of our model to the original DN-CBM using the same hyperparameters ($v_2$). We observe that, under these hyperparameters[3], our fine-tuned model underperforms, with accuracy drops ranging from $-0.7\%$ to $-4.1\%$, depending on the dataset. Although concept interventions remain effective for the fine-tuned model, a noticeably larger drop in accuracy is observed on the Waterbirds-100 dataset, as shown in Table 2. In the concept intervention setup from Rao et al. (2024), only the top five contributing concepts for a given class are used as predictive features, and interventions are performed on these. This setup inherently disadvantages our method. For example, a concept labeled "duck" that represents a broad direction in the feature space associated with birds may serve as a more informative predictor than a neuron that is highly aligned specifically to "duck". Having addressed the accuracy component, we now turn to the interpretability aspect, which we evaluate through our user study.

The results of our user study are presented in Figure 9. Our findings indicate that, across all alignment levels, our model, on average, achieves higher user ratings. This effect is particularly pronounced for low and intermediate-aligned concepts, where the Wilcoxon signed-rank test produces $p$-values of 0.000. This indicates that, even under a conservative significance threshold of 0.1%, the null hypothesis—that there is no average difference in ratings between the reproduced and fine-tuned models—would still be rejected. For the highly aligned concepts, the ratings are similar, and the statistical test yields a $p$-value of 0.176. Notably, the intermediate-aligned concepts receive higher user ratings than the high-aligned concepts in our model. This occurs because enhancing neuron explainability can diminish the meaningfulness of activations, as discussed in Section 3.5

To explain the observed improvements in the user study, we recreate the randomly selected image example from Figure 4 using the fine-tuned SAE in Figure 10. Additionally, we present several example classifications from the fine-tuned model, paired with their corresponding concept-based explanations in Figure 11.

The concepts displayed in Figure 10 are much more semantically coherent with their top activating images in contrast to Figure 4. For instance, the concept "zealand", likely referencing New Zealand, now clearly depicts natural landscapes such as mountains, hills, fjords, and lakes. The concept "silhouette" now reflects dark, shadowy objects, while "commissioned", a broader term often related to commissioned artwork, includes images of jewelry, statues, tables, and paintings. The "sauna" concept is well-represented in the Places365* dataset but is less coherent in other datasets, which contain few or no sauna-related images. Though not perfect, the results show substantial improvement in concept-image consistency.

Figure 11 compares explanations from the original DN-CBM (left) and our fine-tuned model with $C = 10^{-4}$ (right) on randomly selected images from Places365* and CIFAR-10. In the Places365* example, the original model (11a) predicts "Vegetable garden," supported by the concept "ivy," but other concepts like "arnold", "cosmos", "labrador", and "eleven" lack clear relevance. In contrast, our model (11b) predicts "Field wild",

---

[3]Note that these hyperparameters were explicitly optimized for the original DN-CBM, placing our model at a slight disadvantage.

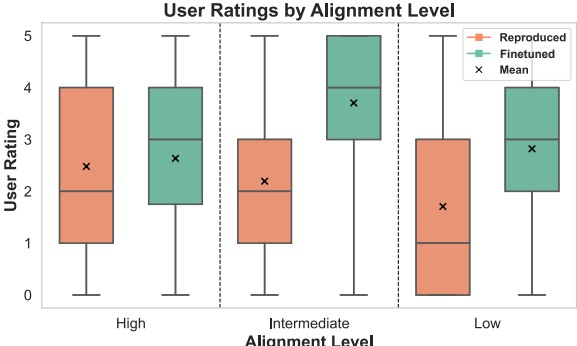

**Figure 9: User study results**. User ratings for the reconstructed DN-CBM (orange) and our model with $C = 10^{-4}$ and probe hyperparameters $v_1$ (green) on Places365*. The ratings are evaluated across three groups based on descending relative cosine similarity scores: high alignment (top 40 highest-aligning concepts of the model), intermediate alignment (40 concepts from the middle), and low alignment (bottom 40 concepts).

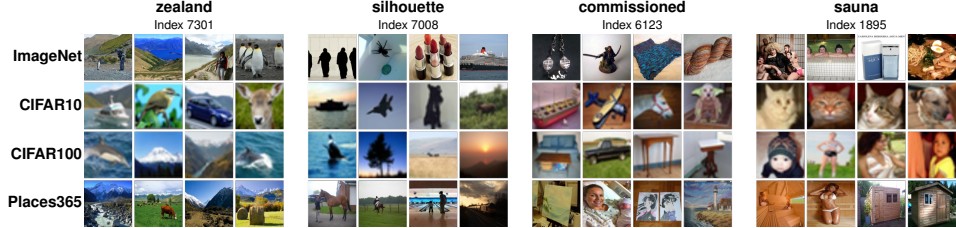

**Figure 10: Task-agnosticity of concept extraction in the fine-tuned model**. We replicate Figure 4 using the fine-tuned SAE ($C = 10^{-4}$), showing top-activating images for named concepts across four datasets.

backed by semantically coherent concepts such as "meadow", "fields", "flower", and "crops". Similarly, for CIFAR-10, the original model (11c) predicts "Horse" but offers unrelated concepts like "pelican", "michigan", "aaliyah", "busty", and "elephants". Our model (11d) instead yields meaningful concepts such as "horses", "equine", and "horseback". These results suggest that our fine-tuned model provides more semantically aligned explanations.

Note that if two latent neurons share the same CLIP label (i.e., their dictionary vectors map to the same label) and both rank among the top 5 contributors, they are merged into a single explanation, making our fine-tuned model (right) appear to produce fewer explanations. Additional examples on CIFAR-100 and ImageNet are shown in Appendix B. While the explanations more accurately reflect the image content in Figure 11, they also tend to align more closely with the class labels, potentially undermining their explanatory value. A similar, though less pronounced, pattern is evident in Figure 5. This issue reflects a fundamental limitation of task-agnostic methods, where predefined features may mirror the class names. We hypothesize that excluding concepts that closely resemble the class name during probing could help mitigate this effect. However, such an approach may come at the cost of reduced predictive accuracy, presenting a trade-off that warrants further investigation in future work.

A key difference between the original SAE and our fine-tuned version lies in how highly aligned concepts are represented. In our model, the concepts with the highest cosine similarity scores tend to be abstract and frequently activated with low magnitude. This likely occurs because they're not tied to specific scene types and are more easily optimized by the loss function. Despite their high alignment, these concepts have low predictive power and rarely influence the final probe decision during inference. As a result, displaying a similar task-agnosticity plot, Figure 10, for the four most aligned concepts in our model isn't meaningful. These abstract concepts (such as "judaism") do not appear in any dataset and these neurons have no impact on inference, thus are not relevant when researching model reasoning. Instead, we focus on the faithfulness of neurons that actually contribute to the model's reasoning.

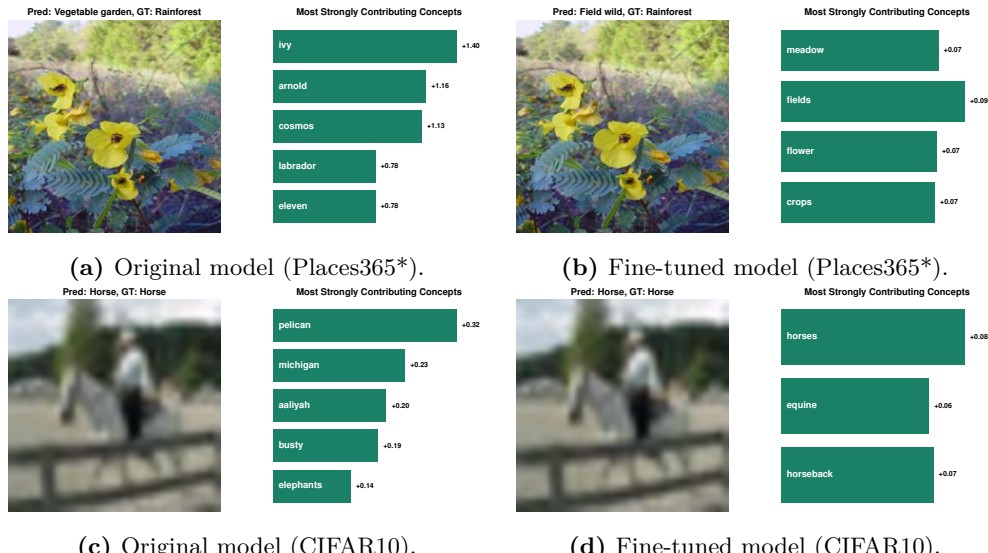

**(a)** Original model (Places365*).

**(b)** Fine-tuned model (Places365*).

**(c)** Original model (CIFAR10).

**(d)** Fine-tuned model (CIFAR10).

**Figure 11: Explaining decisions using DN-CBM and our extension**. The top row shows an example from Places365* with the predicted class, ground truth, and top contributing concepts for both the original DN-CBM (left) and our fine-tuned model with $C = 10^{-4}$ (right). The bottom row presents a similar comparison for CIFAR10.

## 5    Discussion

Our study successfully reproduces **C1** and **C3**, demonstrating that the DN-CBM framework can effectively uncover latent concepts in the data without pre-selecting them while maintaining competitive classification accuracy. Our results for **C2** demonstrate that the authors' work is reproducible; however, the claim is not fully supported, as additional experiments, particularly those involving lower-aligned concepts, often fail to show meaningful contributions to the decision-making process. Fine-tuning the DN-CBM with an extended loss function that drives the dictionary vector of neurons towards explainable CLIP vectors enhances the interpretability of these neurons, as supported by both qualitative and user study analyses. However, this comes at the cost of classification accuracy, introducing a trade-off that can be controlled by adjusting the value of $C$, provided that $C$ does not increase to the point where activations in the hidden state of the SAE diffuse excessively. Future research could involve conducting user studies across a range of values for the cosine penalty parameter $C$.

The concept of an alignment loss through the $C$ penalty term could also be generalized to incorporate different proximity measures. While cosine similarity is one approach to assessing neuron-concept alignment, it may not always be the most effective metric. Determining similarity between vectors in high-dimensional spaces is inherently challenging (Aggarwal et al., 2001).

Lastly, we propose several general directions for future research that are not necessarily specific to our fine-tuned model: removing vague concepts from the vocabulary set, using a larger dataset than CC3M to develop a more general off-the-shelf SAE, discouraging low-aligning and class name mirroring concepts from contributing to classifications through the probe, and introducing a threshold to exclude concepts with low similarity.

**What was easy**. Reproducing the study was relatively straightforward due to the authors' well-documented, publicly available code. The GitHub repository included clear instructions on setting up the environment, running experiments, and reproducing figures, making it easy to verify the majority of the original claims. We considered it unnecessary to contact the original authors.

**What was difficult**. Despite the code being well-documented, we encountered some minor issues, such as errors related to storing results. These were manageable but required some extra debugging. Additionally, the paper did not clearly specify all hyperparameter settings, and while the GitHub code provided some guidance, inconsistencies in argument values added to the difficulty of replicating the results exactly.

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

# A   Additional explanation

### A.1   Survey method

In our user study, we aimed to evaluate concept alignment across different models. Below, we provide additional details on participant selection and survey structure.

**Participant recruitment and survey distribution**. The survey was distributed to a diverse group of participants, including university students, colleagues, friends, and family. To ensure unbiased evaluation, participants were not informed which concepts originated from our model versus the reproduced DN-CBM model.

**Survey structure and model parameters**. The parameter $C$ in our model was set to $10^{-4}$. Before answering the main survey questions, participants were provided with example questions to familiarize them with the task (Figure 12). An example question from the study is included in Figure 13).

UNIVERSITEIT VAN AMSTERDAM

**Example**: we will ask you to judge whether the displayed images align well with the given concept. For each concept, you will see up to five images. Your task is to assign a score from 0 to 5, where **0 means none of the images match the concept**, and **5 means all the images match the concept**, based on your judgment.

Answer: 1

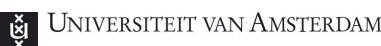

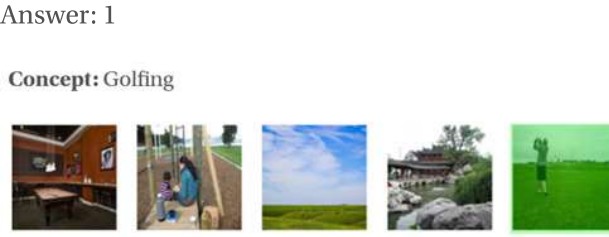

Answer: 5

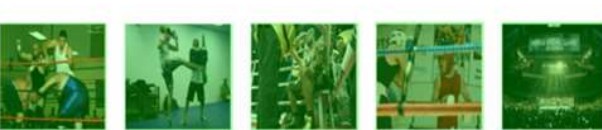

**Figure 12: Examples presented at the start of the survey**. At the start of the survey, we included three sample questions (two of which are shown in this figure) to help participants become familiar with the question format and response process.

**Figure 13: Example of a question in the user study**. We present five randomly selected images in which "Tomb" appears among the top five explaining nodes in the local explanation.

### A.2   Wilcoxon signed-rank test

To determine whether the differences in user ratings between the two models are statistically significant, we employed the Wilcoxon signed-rank test (Wilcoxon, 1945). This test is a non-parametric alternative to the paired $t$-test and is particularly suitable when the assumption of normality is violated.

Let $X_i$ and $Y_i$ represent the average user rating given by participant $i$ for the reproduced DN-CBM and our model respectively, at a specific alignment level (high, intermediate, or low). The difference in ratings for participant $i$ is:

$$D_i = Y_i - X_i, \tag{8}$$

where $D_i$ represents whether the participant preferred one model over the other for that alignment level.

For each alignment level, we test:

**Null hypothesis ($H_0$).** The average difference in ratings across participants is zero ($\overline{D} = 0$). This implies that there is no significant preference for either model.

**Alternative hypothesis ($H_A$).** The average difference is not zero ($\overline{D} \neq 0$), meaning participants systematically rate one model higher.

**Assumptions**. The Wilcoxon Signed-Rank Test assumes that the two samples are dependent, meaning the data consists of paired samples. Moreover, the distribution of $D_i$ should be approximately symmetric around the median. Lastly, the test requires that the data is ordinal. These assumptions hold for our data, as the ratings range from 0 to 5, and each participant's ratings are dependent.

# B Additional qualitative results

## B.1 Generalization local explanation

This section provides an extra local explanation for the reproduced DN-CBM and examines the local explanation for ImageNet and CIFAR10.

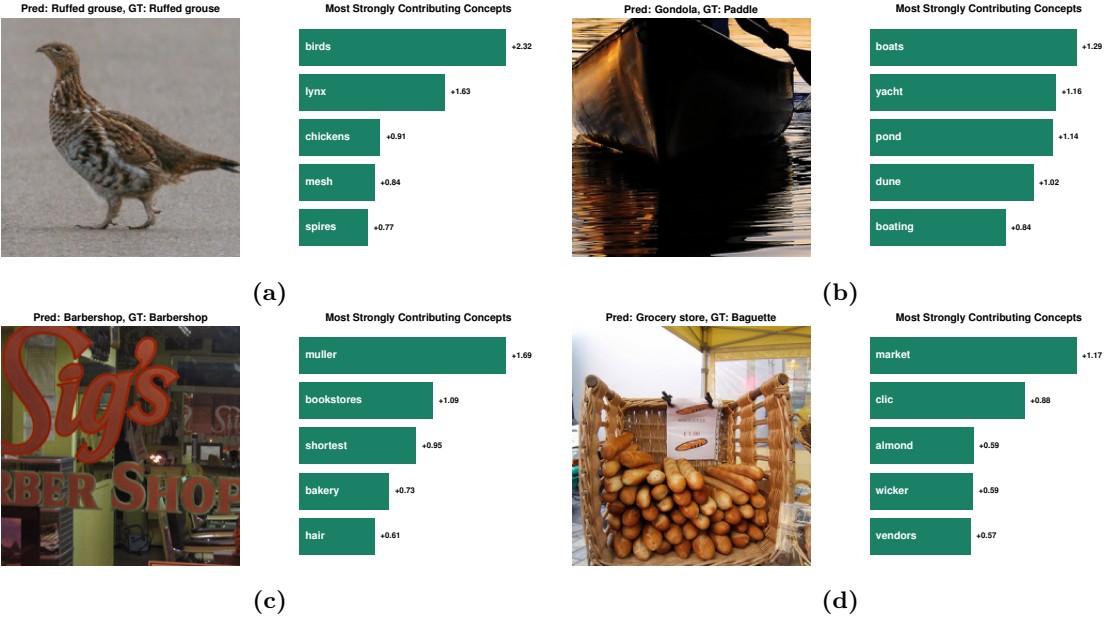

**Figure 14: Explaining decisions using the reproduced DN-CBM**. We present randomly drawn examples of images from the ImageNet dataset alongside the top concepts contributing to their classification. Figures 14a and 14c are correctly classified, whereas Figures 14b and 14d deviate from the ground truth labels. While the predicted labels for Figure 14b appear reasonable, and most labels for Figure 14a are also interpretable—except for the label "lynx"—the classifications for Figures 14c and 14d are less coherent. Specifically, concepts such as "muller", "bookstores", "shortest", and "bakery" fail to provide a meaningful rationale for the barbershop classification. Furthermore, in Figure 14d, one of the highest contributing concepts is "clic", which lacks a clear semantic interpretation, making it difficult to understand its role in the model's decision-making process.

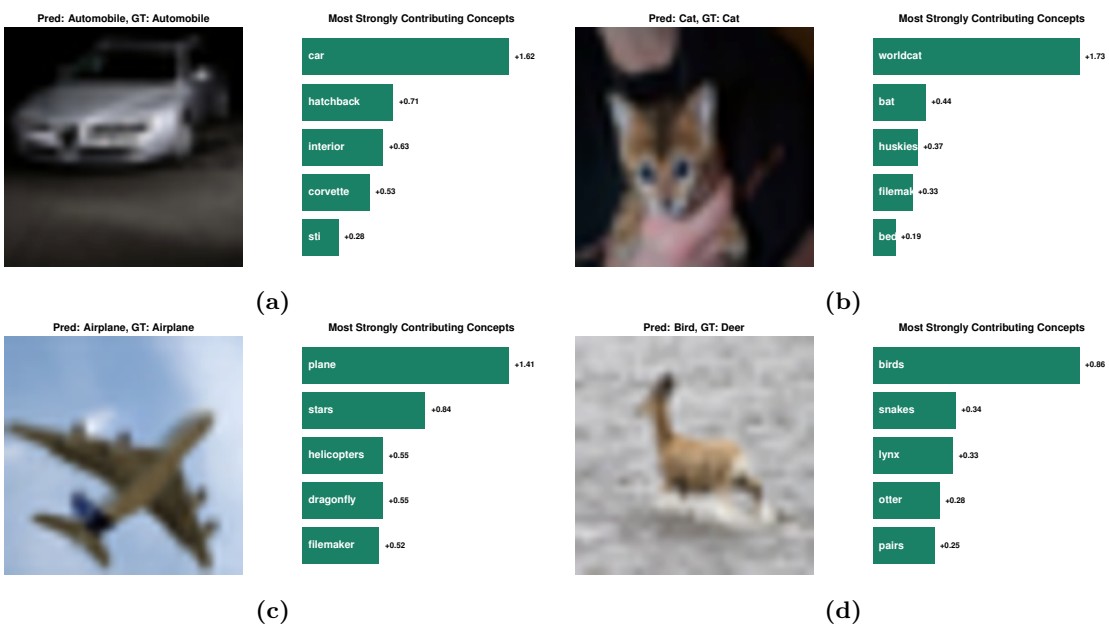

**Figure 15: Explaining decisions using the reproduced DN-CBM.** We present randomly drawn examples of images from the CIFAR10 dataset alongside the top concepts contributing to their classification. Our observations indicate that Figures 15a, 15b, and 15c are correctly classified, while Figure 15d is misclassified. The explanations for the classifications of "Automobile", "Airplane", and "Cat" appear reasonable, as they include relevant concepts such as "car", "plane", and "worldcat." In the case of Figure 15d, which is misclassified as a "Bird", the contribution of the "birds" node can be interpreted as a plausible factor. However, the inclusion of concepts such as "snakes", "lynx", and "otter" is less intuitive and does not provide a clear rationale for the model's prediction.

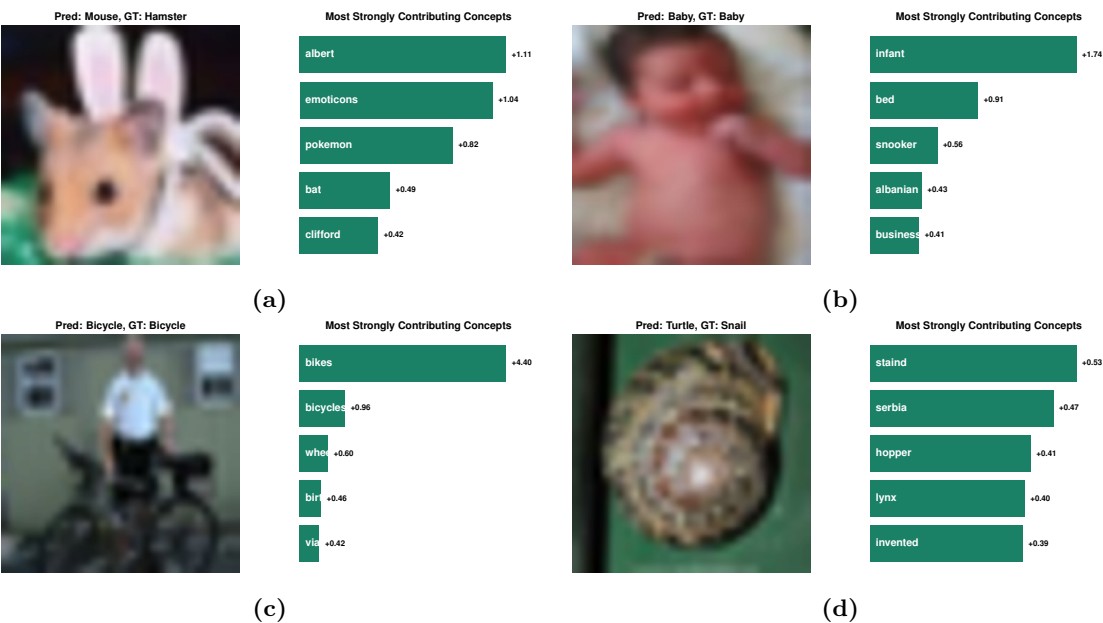

**Figure 16: Explaining decisions using the reproduced DN-CBM.** We present randomly drawn examples of images from the CIFAR100 dataset alongside the top concepts contributing to their classification. Our analysis reveals that Figures 16b and 16c are correctly classified, whereas Figures 16a and 16d are misclassified. The primary contributing concepts for the classifications of "Bicycle" and "Baby", such as "bikes" and "infant", provide meaningful and interpretable justifications. However, in the case of the misclassification as "Mouse," the concepts "albert", "emoticons", and "pokemon" do not offer a coherent explanation for the model's decision. Similarly, for the misclassification as "Turtle," the contributing concepts "staind" (a band), "Serbia", and "hopper" lack clear semantic relevance, making the prediction difficult to interpret.

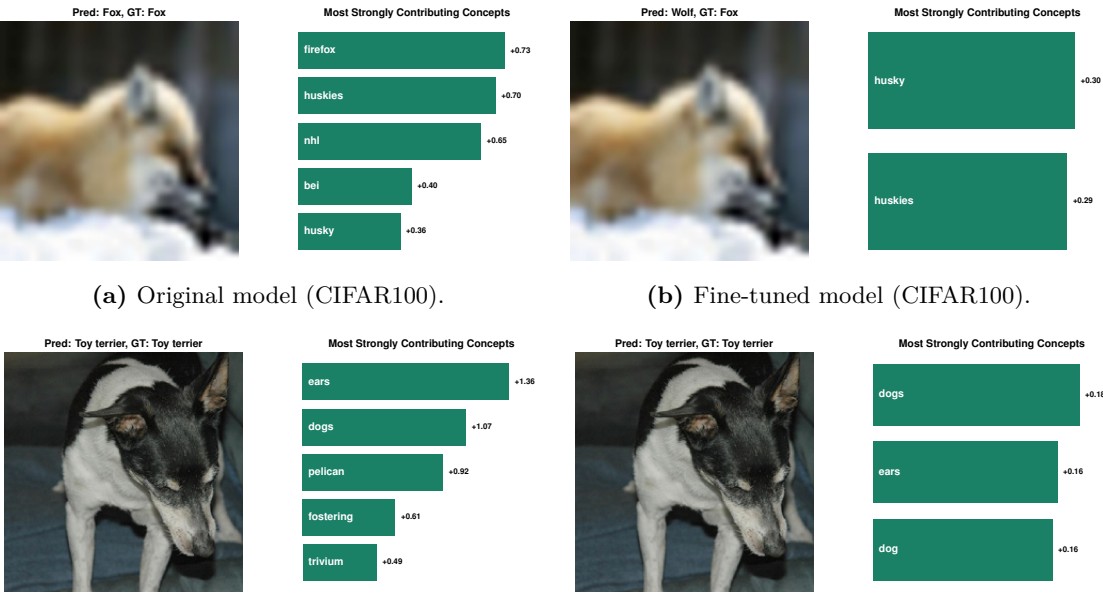

**(a)** Original model (CIFAR100).

**(b)** Fine-tuned model (CIFAR100).

**(c)** Original model (ImageNet).

**(d)** Fine-tuned model (ImageNet).

**Figure 17: Explaining decisions using DN-CBM and our extension**. The top row shows an example from CIFAR100 with the predicted class, ground truth, and top contributing concepts for both the original DN-CBM (left) and our fine-tuned model with $C = 10^{-4}$ (right). It is observed that our model predicts the incorrect label "wolf", yet the rationale behind this classification is interpretable, with contributing concepts such as "husky" and "huskies". Given that CIFAR100 does not contain a "husky" class, "wolf" is identified as the next closest match. In contrast, the original model correctly predicts the label, with explainable concepts such as "firefox", "husky", and "huskies", although concepts like "nhl" and "bei" are less interpretable. The bottom row compares similar examples for ImageNet. In this case, both models correctly classify the image as "Toy terrier". Our model provides a set of fully interpretable concepts, including "dogs", "ears", and "dog". The original model also identifies several explainable concepts such as "ears", "dogs", and "fostering", but includes less interpretable concepts like "pelican" and "trivium".

## C  Additional quantitative results

### C.1  Concept interventions

For examining the effectiveness of concept interventions on both the original DN-CBM and our proposed fine-tuned method, we use the same methodology as Rao et al. (2024): We use the SAE for the CLIP ResNet-50 model, and train a linear probe for the binary classification task of the Waterbirds-100 dataset. For increased sparsity, we prune the weights for each class, leaving only the five largest weights and replacing the rest with zeroes.

For each class, we then look at the five largest weights with their appropriate names, and classify them as a 'Bird Concept' or a 'Non-Bird Concept'. In Table 4 and 5, we show the five largest retained weights (concepts) per class (landbird or waterbird) for the reproduced model and our fine-tuned model respectively. For our fine-tuned model, we used a hyperparameter value of $C = 10^{-4}$ and removed duplicates in the five largest weights. Just like Rao et al. (2024), we perform two sets of interventions: (1) keeping only the bird concepts, and (2) removing only the bird concepts.

| Class | Bird Concepts | Non-Bird Concepts |
|---|---|---|
| Landbird | sparrow, parrot | branch, wetland, woodland |
| Waterbird | ibis, duck | pond, landing, beach |

**Table 4:** Set of concepts in our evaluation for performing interventions on the original reproduced DN-CBM. For each class, we manually classify each of the five concepts as being a bird or non-bird concept, for applying appropriate interventions.

| Class | Bird Concepts | Non-Bird Concepts |
|---|---|---|
| Landbird | robin, sparrow | woods, jungle, woodland |
| Waterbird | swan, duck | ocean, beach, water |

**Table 5:** Set of concepts in our evaluation for performing interventions on our fine-tuned DN-CBM. For each class, we manually classify each of the five concepts as being a bird or non-bird concept, for applying appropriate interventions.

## C.2 Cosine similarity score distributions

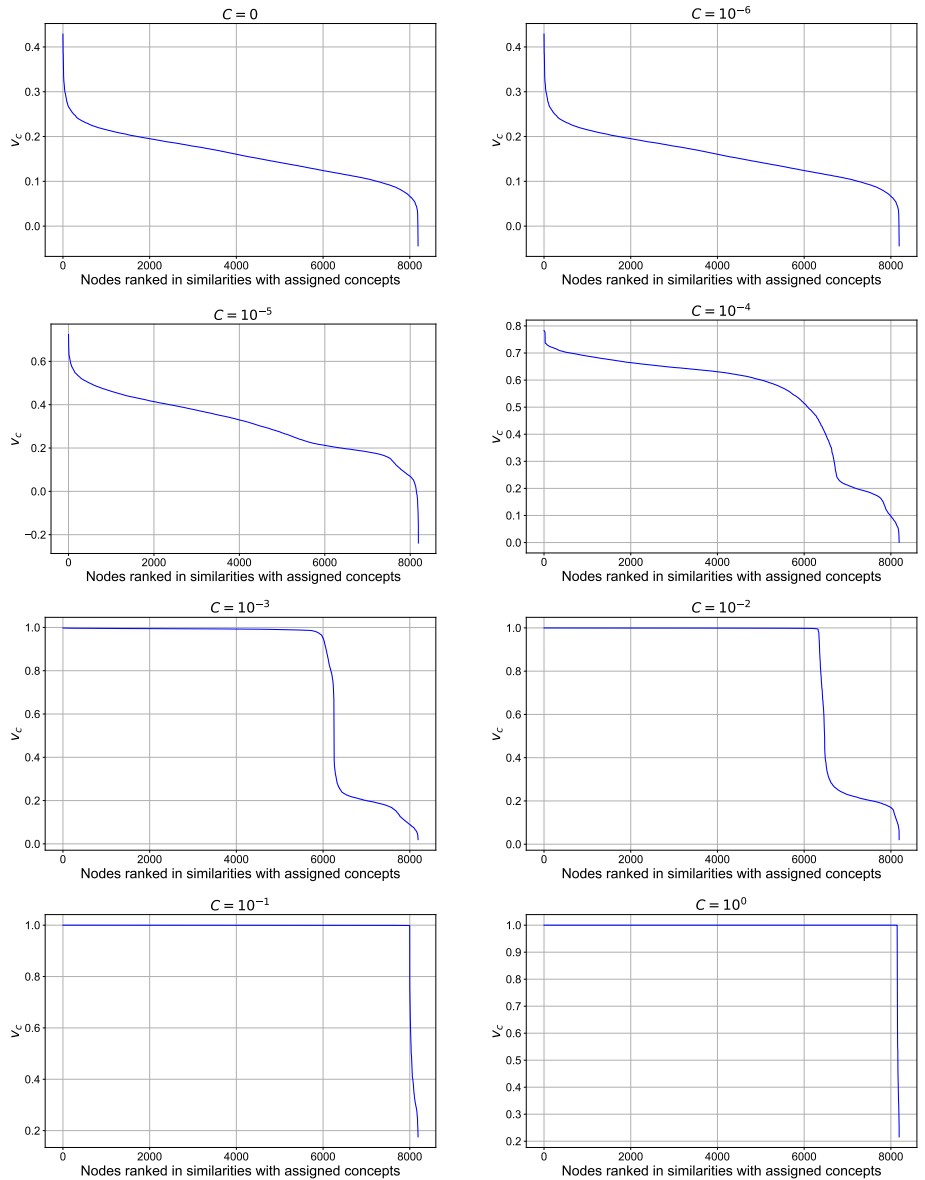

**Figure 18: Cosine similarity distribution for different parameters** $C$. The ranked cosine similarity values of the assigned concepts after fine-tuning with varying penalty parameters are presented. $C = 10^{-6}$ yields a similar distribution to the original cosine distribution of $C = 0$. For $C = 10^{-5}$, the shape of the distribution of cosine scores remains similar to that for 0, but with a substantially larger range. When $C = 10^{-4}$, a noticeable shift in the distribution emerges, with most similarity values becoming positive and a larger proportion of concepts exhibiting higher cosine similarity. For $C = 10^{-3}$ and $C = 10^{-2}$, the cosine similarity scores reach 1 for the first 6,000 nodes before rapidly declining to 0.2 or lower. For $C = 10^{-1}$ and $C = 10^{0}$, the cosine similarity score goes to 0.2 after 8000 nodes.

