# OpenReview forum: "Revisiting Discover-then-Name Concept Bottleneck Models: A Reproducibility Study"
_TMLR — Accepted by TMLR_

### Review · Reviewer_XTw2 · 2025-03-12

**Summary Of Contributions:**

The paper reproduces the results of Rao et al. (2024) along the lines of three central claims: (C1) automated concept discovery, (C2) interpretability and (C3) performance. The findings of the paper suggest that (C1) and (C3) can be verified whereas some questions regarding (C2) arise. Further, the authors propose an extension by explicitly regularizing the concept embedding space based on pre-defined concepts. They presents results showing that this improves the concept coherence at the price of reduced predictive accuracy.

**Audience:**

Yes

**Broader Impact Concerns:**

Not applicable.

**Claims And Evidence:**

Yes

**Requested Changes:**

More details around C2 as outlined above. A fully conclusive solution to this problem is out of scope but some effort should be put into improving the manuscript in this regard. In particular, this central findings should more prominently mentioned and the resulting limitations for DN-CBM further discussed. For example consider mentioning this finding also in the abstract.

They present a distribution of concept similarity for the proposed extension in appendix Fig. 16. It might be worth moving sub-figure C=10^-4 into Fig. 2 in the main part. This would enable to compare it to the original methodology and provide more intuition on the effect of the regularization. I leave this up to the authors to decide.

**Strengths And Weaknesses:**

# Strenghts
The writing style and presentation is clear. All experiments seem to be technically well performed and the code is provided via a GitHub repository. Remarkably, the authors have also independently verified the results of a user study presented in  Rao et al. (2024). Overall, the paper gives a good overview of the capability of the DN-CBM method.

# Weaknesses
The problems around (C2, interpretability) are not further discussed. This is a missed opportunity since faithful interpretability is core for any method in XAI. In particular also the proposed extension does not address this point successfully (see discussion below).
The paper closely resembles the original paper. While this is necessary for a reproducibility study, some well-designed additional investigation of DN-CBM would help to judge the robustness of the original findings. In particular, regarding the interpretability claim C2.

The paper nicely highlights the problems around low-aligned concepts in Fig. 4. However, the authors lack to discuss this phenomenon, its consequences and where it originates from. More details how the problem of incoherent concepts originates from the underlying methodology would be helpful for the community. For example, why is it not possible to introduce a threshold and exclude concepts with low similarity from the final CBM model?

Complementary, I spotted concepts names which are very close to the predicted class (Fig. 9d, GT: Horse with concept horses&horseback) or nearly identical (Fig. 15, eg. Huskies&husky or dogs & dog). This seems to be a general issue of the DN-CBM methodology as it is also present in the extended method. This has the potential to severely limit the usefulness in practice. Identifying a concept with the same name as the predicted class does not provide new information about the model reasoning. Thus, this deserves further investigated. One possible idea would be to modulate the size/capacity of the concept vocabulary excluding too similar names.

Lastly, the proposed extension (based on regularization) seems to be overly restrictive. It has significant impact on the performance of the CBM model even though this is major selling point of the original methodology. Also the resulting concepts are very narrow and closely related to the original class. This raises questions regarding the usefulness of this extension in practice.

---

> ### Author Response · Authors · 2025-04-21
> **Thanks and response to concerns**
>
> Dear reviewer XTw2,
>
> We appreciate your response and the constructive feedback on our work. In what follows, we provide a point-by-point reply to your comments, along with the concrete changes we have made as a result. For each point, we include a brief summary of your feedback to ensure that our understanding is aligned. If anything in our response is unclear, please don’t hesitate to reach out.
>
> >Point 1:
> The reviewer points out that the overarching problems around (C2, interpretability) is not discussed. This problem refers to the faithfulness of the interpretability of the DN-CBM. Furthermore the reviewer points out that the proposed extension does not address this point successfully as outlined in point 3. The review points out that some additional investigation of the DN-CBM to judge its robustness is missing but would be useful.
>
> We strongly agree that faithfulness of interpretability is essential in  XAI, and that the challenges associated with C2 therefore deserve a thorough and dedicated discussion. In fact, this is precisely the direction we aimed to focus on in our work. Besides reproducing, we additionally investigated the distribution of the cosine similarities (Fig. 2), which identifies the problem of faithfulness as some concepts turn out to be better aligned than others. Furthermore, our study includes a survey with 200 respondents that essentially evaluates the faithfulness or interpretability of the DN-CBM (for both the original and our extended version of the CBM). Finally, our extension is specifically designed to make the model explanations more faithful.
>
> >Point 2:
> The reviewer appreciates our illustration of low-aligned concepts (Fig. 4), but points out that we do not adequately discuss the origin and implications for this issue, which would be helpful for the community. Furthermore, the reviewer asks why low-similarity concepts cannot simply be excluded using a threshold.
>
> To validate C2, we conducted a cosine similarity analysis (Fig. 2) between dictionary vectors and their ground-truth CLIP embeddings. We find that the cosine scores range broadly from approximately -0.01 to 0.42, indicating the presence of lower-aligned concepts. This figure serves as the explanation of the origin of the problem and forms a natural bridge and motivation to further investigation of lower aligned concepts (Fig. 4), and ultimately serves as the motivation for our model extension.
>
> We agree that our work could benefit from more clarification on this topic. Therefore, in the new iteration of our submission we added explicitly that Fig. 2 raises questions on the faithfulness of the interpretability in Section 4.1.
>
> Furthermore, we discussed the more fundamental origin of the problem in the start of Section 3.4.2. (in the new iteration of our paper this section is the start of Section 3.5). In this section we outline that the problem of faithfulness stems from the post hoc approach of the DN-CBM.  Specifically we would like to refer to the part where we write the following about the DN-CBM method:  *“This approach is compelling for maintaining the model's accuracy, as it does not constrain the model to be inherently explainable during training. However, the degree of explainability is theoretically limited as certain aspects of a freely trained model cannot be fully captured by a single word or concept due to the constraints of human language.”*
> In this paragraph we try to capture and discuss the more fundamental issue with the DN-CBM method, that leads to Fig. 2 and therefore Fig. 4.
>
> As for a discussion of the implications of this issue: since explanation quality is subjective, we chose not to explicitly claim that DN-CBM is unfaithful or detail its consequences. Instead, we presented supporting evidence—through images and a user study—to allow readers to draw their own conclusions, and addressed the issue through our model extension. Furthermore, we felt it unnecessary to elaborate on the general importance of explanation faithfulness, as this is an established and widely accepted goal within the XAI community.
>
> > why is it not possible to introduce a threshold and exclude concepts with low similarity from the final CBM model?
>
> We agree this could be a valuable extension to DN-CBM and have considered it. We opted against it due to two challenges: (1) Applying a hard cutoff would likely reduce model accuracy severely. (2) Defining a meaningful lower bound for concept alignment is complex and subjective, as defining "good" alignment in a high-dimensional space is challenging. We find it more natural and robust to focus on improving the overall cosine similarity scores. This avoids the need for threshold tuning and encourages better concept quality across the board. We agree that this is an interesting line of research, and we will acknowledge it in the future work section of our revised submission (Section 5).
>
> We will continue this comment in the next block.

---

> > ### Author Response · Authors · 2025-04-21
> > **Thanks and response to concerns (part 2)**
> >
> > > The reviewer notes that concept names closely resembling the target class labels may limit the explanatory value of the DN-CBM, as they do not offer additional insight into the model’s internal reasoning. This seems to be a general issue of the DN-CBM methodology.
> >
> > Thank you for addressing this point, in the revised version of our work we highlight this general limitation of the DN-CBM framework (section 4.1 along with figure 5).
> >
> > This could indeed be considered a more general limitation of the DN-CBM (or any CBM) approach. Explaining the classification of a horse with “the neuron for horse is activated”, shifts the question towards understanding why that particular neuron responds to the concept of a horse. However, this is not a problem unique to cases where the explanation label overlaps with the class label. If the model identifies a horse through a neuron labeled “tail,” one could similarly ask: how does the model recognize the tail? This highlights a broader, more fundamental challenge in CBMs, not limited to class-aligned explanations. Therefore, we argue that model explanations based on a class label are no less insightful for understanding the model’s reasoning than any other label.
> > Addressing this issue is not within the scope of our work, instead, we focus on reproducing and improving the model interpretability within its framework. In the DN-CBM context, if a model has a neuron labeled “horse,” it is desirable for it to activate for an image containing a horse, as this indicates that the neuron is labelled (/interpreted) appropriately. Our experiments show that, in the original model, this neuron does not activate for such images. However, after implementing our fine-tuning approach, this neuron does.
> >
> > > The reviewer raises concerns that our proposed extension significantly impacts the accuracy of DN-CBM, which is arguably a key strength of the original DN-CBM method. This, in turn, raises questions about the overall usefulness of the extension.
> >
> > We agree that the extension is restrictive and would like to clarify its intended use case.
> >
> > The primary goal of our extension is to explore the trade-off between interpretability and accuracy within the DN-CBM framework.
> >
> > While DN-CBM offers a degree of explainability, it is not explicitly trained with interpretability constraints. This allows it to achieve high accuracy, unlike other CBMs. However, in sensitive domains (e.g. medical), faithful interpretability becomes critical, and relying on explanations that may not align with the model’s true reasoning can be problematic.
> > Our extension addresses this issue in two ways:
> > 1. It highlights the challenge of faithful interpretability (C2) in the existing DN-CBM, which we believe has not been sufficiently discussed in the original work.
> > 2. It introduces a mechanism that enables users to actively balance accuracy and interpretability by tuning the hyperparameter C. This allows for model adjustments based on the downstream task without significantly altering the DN-CBM architecture.
> >
> > While we do not claim to have fully solved this problem, we hope our work represents a meaningful step toward more robust and user-controllable interpretability in CBMs.
> >
> > > The reviewer asks for a more inclusive discussion on the limitations of the DN-CBM framework. Furthermore finds that the central finding of this limitation on C2 should be more prominently discussed, for example, mentioned in the abstract.
> >
> >  The lack of faithfulness of the DN-CBM should indeed have been explicitly addressed. We have added it to the abstract in the revised submission and in section 3.5.  We have not included a more inclusive discussion of this limitation since we feel like the main message of our paper already addresses this limitation.
> >
> > > It might be worth moving sub-figure C=10^-4 into Fig. 2 in the main part.
> >
> > We agree with this suggestion and have moved this sub-figure to Figure 8, along with subfigure $C=10^{-3}$.

---

> > > ### Comment · Reviewer_XTw2 · 2025-04-22
> > > **Thank you for your reply**
> > >
> > > Dear authors,
> > > Thank you for responding to the review. From my perspective, the manuscript has generally improved.
> > >
> > > I appreciate that you explicitly mentioned the matching of concepts and class labels as a general limitation around Fig. 5.
> > > As this problem is more severely exposed in Fig. 11, you might consider providing a reference to these examples here.
> > >
> > > You state that the alignment ranges from a cosine similarity of -0.01 to 0.42. On this basis you argue that it is difficult to introduce a cut-off. However, a cosine similarity of zero indicates two independent directions. How do you justify this within your framework?
> > > Consequently, would it not be more faithful to exclude these concepts? This would allow to explicitly measure the trade-off between interpretability (alligned concept and unexplained feature space (leading to reduced performance).
> > > If you have any fruitful thought on this, this might help guide future research.

---

> > > > ### Author Response · Authors · 2025-04-24
> > > > **Follow up response**
> > > >
> > > > Thank you for the thoughtful follow-up comments. We address the points raised below with further clarification and revisions.
> > > >
> > > > - On the matching of concepts and class labels:
> > > >
> > > > We agree that a more detailed explanation of this issue should accompany Figure 11. In the latest iteration of our work, we include the following text:
> > > >
> > > > "While the explanations more accurately reflect the image content in Figure 11, they also tend to align more closely with the class labels, potentially undermining their explanatory value. A similar, though less pronounced, pattern is evident in Figure 5. This issue reflects a fundamental limitation of task-agnostic methods, where predefined features may mirror the class names. We hypothesize that excluding concepts that closely resemble the class name during probing could help mitigate this effect. However, such an approach may come at the cost of reduced predictive accuracy, presenting a trade-off that warrants further investigation in future work."
> > > >
> > > > We also propose discouraging class name mirroring concepts from contributing to classifications through the probe as a direction for future work in the discussion.
> > > >
> > > > - On the cut-off
> > > >
> > > > While the intuition of excluding concepts with a cosine similarity below zero to achieve a more faithful model is sound and offers a seemingly universal cutoff point, applying it directly within the current DN-CBM framework has limited effect. Our analysis reveals that only a small fraction of latent neurons exhibit negative alignment in the reproduced DN-CBM (4 out of 8192 hidden dimensions, to be precise). Consequently, this exclusion alone would not resolve the core explainability issues we observe.
> > > >
> > > > As Figure 4 illustrates, concepts with a positive cosine similarity of approximately 0.1 can exhibit top-activating images that lack a clear semantic connection to the concept itself. Conversely, Figure 3 demonstrates that similarities in the 0.3–0.4 range correspond to readily interpretable visual features. This discrepancy naturally raises a critical question: what constitutes an appropriate threshold? Defining such a threshold within the original DN-CBM framework presents two significant challenges:
> > > >
> > > > Firstly, selecting any specific numerical threshold remains inherently subjective. While examining top-activating images for concepts with increasing cosine similarity offers some guidance, pinpointing a definitive cutoff point lacks an objective basis.
> > > >
> > > > Secondly, even a seemingly conservative threshold of 0.1, motivated by the inconsistencies depicted in Figure 4, would necessitate the exclusion of approximately 2000 out of 8192 hidden dimensions in the SAE's hidden state. Such a significant reduction in the feature space would lead to a substantial and undesirable decline in model accuracy.
> > > >
> > > > Considering these limitations, we believe that shifting the entire distribution of cosine similarities towards higher values represents a more principled and potentially less damaging strategy for enhancing interpretability.
> > > >
> > > > We hope this response addresses the concerns raised. If there are any remaining questions, we would be happy to clarify further.

---

### Review · Reviewer_RjVw · 2025-03-19

**Summary Of Contributions:**

This paper reports on a reproducing experiment of DN-CBM, which was presented at ECCV 2024. The paper investigates DN-CBM in three perspectives: (i) the ability of DN-CBM to automatically discover concepts, (ii) the interpretability, and (iii) the performance. While the paper confirms the reproducibility of (i) and (iii), it reports potential issues related to interpretability that were not discussed in the original paper. Specifically, it points out that the automatically discovered concept dictionary vectors extracted by sparse auto-encoder (SAE) cannot accurately provide interpretability for input images when the cosine similarity with the original textual concept vectors is low. To solve this issue, the paper proposes fine-tuning the SAE dictionary vectors by regularizing them to align with specific text concepts. The proposed method confirms that even concepts with relatively low alignment have high interpretability for humans.

**Audience:**

Yes

**Broader Impact Concerns:**

Nothing to report.

**Claims And Evidence:**

Yes

**Requested Changes:**

- Please clarify the motivation for re-evaluating DN-CBM (see W1).
- DN-CBM limits the number of explainable concepts to the dimension $h$ of the SAE dictionary matrix. The paper should add a discussion of what impact scaling $h$ has on interpretability and performance.
- Please evaluate human intervention. In particular, what changes do the proposed methods make to the intervention results?
- Ultimately, DN-CBM may be able to achieve full interpretability by replacing the SAE dictionary vectors with the corresponding text concept vectors. Is it possible that static replacement may produce better interpretability results than applying the proposed method? I would recommend adding this discussion.

**Strengths And Weaknesses:**

### Strengths
+ **S1.** The paper carefully designs reproducing experiments and evaluates DN-CBM from various perspectives, including human evaluation.
+ **S1.** The paper points out issues with the interpretability of DN-CBM through case studies and user studies. Since DN-CBM forcibly maps SAE dictionary vectors to text concepts, it is reasonable that the interpretability of concepts with low alignment is not good.
+ **S1.** The paper proposes fine-tuning the SAE dictionary vectors by regularization to align them with specific text concepts in order to improve the interpretability of DN-CBM. This strengthens the mapping between dictionary vectors and specific text concepts, improving interpretability but degrading performance.
### Weaknesses
- **W1.** The paper does not show strong motivation for the need to reproduce the DN-CBM experiments. The DN-CBM is one of many CBMs that are still being developed, and it is unclear why the DN-CBM in particular should be the focus of attention. The goals of the verification addressed by the paper and the new knowledge advancement it provides to the community are unclear.
- **W2.** The experimental facts provided by the paper are mostly trivial, and the differences are small. The paper discusses the fact that the claim (ii) cannot be partially reproduced. However, it seems to point out the issues that arise when re-evaluating DN-CBM from a different perspective (e.g., low alignment concepts or large sample size) rather than the original paper is not reproducible. In addition, the fact that the SAE dictionary vectors are forcibly mapped to a finite number of concepts is a clear limitation of the DN-CBM design, and it is theoretically obvious that this can not provide a perfect interpretation. From this perspective, the proposed extension method emphasizes this limitation, suggesting that DN-CBM, which has a trade-off between interpretability and performance, is impractical.
- **W3.** The paper arbitrarily restricts the content of the reproducing experiments. In particular, human intervention is an important aspect when evaluating CBM, and it is unnatural not to re-evaluate this. The justification for excluding a re-evaluation of human intervention is lacking because the verification goal addressed by the paper is unclear as mentioned in W1.

---

> ### Author Response · Authors · 2025-04-21
> **Thanks and response to concerns and requested changes**
>
> Dear reviewer RjVw,
>
> Thank you for your thoughtful and constructive review of our paper. Below, we provide a point-by-point response to your comments, along with the corresponding changes made in the revised version. We also summarize each point briefly to ensure that we have accurately understood your concerns. If anything remains unclear, we would be happy to provide further elaboration.
>
> > Please clarify the motivation for re-evaluating DN-CBM (see W1).
>
>  In the revised version, we have expanded the Introduction to better articulate our motivation. In particular, we now emphasize that DN-CBM is a recent and high-performing concept bottleneck model that achieves strong accuracy while claiming interpretability in a task-agnostic manner. These qualities make it both influential and methodologically novel.
>
> > The paper should discuss how scaling the size of $h$ affects both interpretability and performance.
>
> Thank you for highlighting this point. We agree that the dictionary size $h$ is a central parameter that affects both performance and interpretability. In response, we conducted additional experiments analyzing how varying h influences alignment and accuracy on our reduced Places365 dataset. With the original value of $h$ (8192), we achieved an accuracy of 50.0%. Increasing $h$ to 1.5x the original size reduced accuracy to 49.6%, but slightly improved concept alignment. Increasing $h$ further to 2x the original size decreased accuracy to 48.7%, while alignment slightly increased again. This suggests a trade-off: a larger $h$ value allows finer-grained alignment but comes at the cost of performance.
>
> We found that scaling $h$ shifts the mean of the cosine similarity distribution slightly upwards, but not as much as our proposed extension does. Additionally, many nodes got assigned the same concepts, training time significantly increased and the performance drop was more significant compared to our proposed method. Therefore, we opted not to include the full analysis in the main text, also due to space constraints and the scope of our research. We hope we have clarified this point enough, please reach out to us if you have any further questions regarding this point.
>
> > Evaluation of human intervention and its changes under fine-tuning (W3): The original DN-CBM study evaluated human intervention, but this paper does not. The omission is unjustified. Additionally, how does the proposed fine-tuning method affect intervention?
>
> In the revised version, we have included a human intervention analysis for both the original and fine-tuned DN-CBM models. These results are presented in Table 2 and align well with the results reported by Rao et al. As for our fine-tuned model, we observe an accuracy drop which we attribute to the fact that our concepts are better aligned, making activations more specific; we keep only four bird-related concepts and our model incentivises these nodes to only activate when those concepts really appear. Despite this accuracy drop, human intervention still proved to be effective.
>
> > Static replacement of SAE vectors with text concept vectors: It is worth discussing whether directly replacing SAE dictionary vectors with text concept vectors could lead to better interpretability than the proposed method.
>
> We agree that this is an important point. However, this suggestion represents an extension beyond the scope of our current work. That said, we did consider this approach during our development and would like to share our thought process:
>
> A natural alternative for exploring the far end of the interpretability and accuracy trade-off, without depending on the extended loss function, would be to directly substitute the SAE dictionary vectors with the corresponding text concept vectors. Such a model could be seen as a maximally interpretable CBM, although it would place strict constraints on the dictionary vectors. This would likely lead to a drop in accuracy due to what we consider an unnecessarily rigid constraint. While a concept can be represented by a specific vector in CLIP space, it is more accurately described as occupying a subregion around that vector. This is because a different set of images, or a different group of annotators, would likely produce a slightly different embedding. Forcing the model’s representations to fall arbitrarily close to a single point within this region may result in a suboptimal balance, potentially compromising accuracy for only modest improvements in interpretability. A more balanced approach would guide the model toward this ideal point while carefully considering the associated accuracy cost, providing a principled compromise between interpretability and performance. This is exactly the goal of our introduced cosine similarity regularizer.
>
> We will continue the last part of our response in the next comment.

---

> > ### Author Response · Authors · 2025-04-21
> > **Thanks and response to concerns and requested changes (part 2)**
> >
> > > Clarification on the reproducibility of Claim 2 (W2): The paper claims that C2 is not fully reproducible, but this is only due to additional evaluation methods not present in the original paper. Therefore, it should not be framed as a failure of reproducibility.
> >
> > We agree that this may be a harsh statement. In the revised version of our work, we adopt a more nuanced approach. To clarify the original intent: In their work, Roa et al. claim explainability  C2, which is partially supported by displaying samples of figures alongside local explanations. Specifically, we believe that reproducing the quality of the local explanations shown in Figures 7, 8, D5, D6, and D7 is not feasible through random sampling of figures. The combination of these figures displays a degree of faithfulness in explanations that we were unable to match by sampling random figures.
> >
> > That said, Rao et al. never claimed to have sampled the images for these local explanations randomly, nor do they provide a detailed discussion of their method. Furthermore, this difficulty does not discredit the proof of concept presented in their work. And more importantly, this cannot be derived from Figure 4 of our work. Therefore, we agree with the suggestion and will rephrase our claim to reflect this more carefully.
> >
> > Once again, thank you for your detailed review. We hope to have answered all of your questions and concerns. Please don't hesitate to reach out for any further clarification.

---

### Review · Reviewer_UByM · 2025-04-08

**Summary Of Contributions:**

This work studies the replicability of Discover-then-Name CBMs (DN-CBMs) by Rao et al. (2024). In particular, it investigates whether DN-CBMs are (1) capturing meaningful concepts without any domain knowledge, (2) producing interpretable predictive models, and (3) producing accurate predictive models. By recreating most of the key experiments of Rao et al., as well as running a larger-scale user study than that done by Rao et al., this work concludes that most of the experimental claims in the work by Rao et al. do hold up to scrutiny. Nevertheless, this paper shows that, contrary to what is discussed in the work of Rao et al., there are still latent dimensions in DN-CBMs that are not fully aligned to meaningful concepts. To address this, this paper proposes a simple modification to the learning objective of Rao et al. that encourages better alignment for all discovered latent dimensions through a fine-tuning process.

**Audience:**

Yes

**Broader Impact Concerns:**

I do not believe there are any broader ethical implications from this work that would merit a Broader Impact Statement.

**Claims And Evidence:**

Yes

**Requested Changes:**

Below, I list some potential changes for each section and their importance in securing my recommendation for acceptance. If time is limited, please focus on all **critical** requests, followed by all **major** requests. If I misunderstood something at any point, just let me know, as this is always possible (apologies in advance if that’s the case!).

### Section 1 (Introduction)

- **(Minor, Attribution)** I would recommend properly citing a work the first time it is introduced. So, when first discussing CBMs in the introduction, please citer them in the same line.
- **(Minor, Attribution)** As above, it would be beneficial if, when discussing “recent innovations” in the introduction, you provide valid evidence and citations of which “innovations” you are referring to.
- **(Critical, Wrong Attribution)** The work of Margeloui et al. cited in the introduction does not, in any form of fashion, discuss issues of using LLMs for CBMs. They show some concerning results on CBMs in general (in their instance, they use ground-truth concept annotations rather than LLM-generated annotations). Using this work to say that “LLMs can lead to unfaithfulness to the model’s reasoning process” is misleading and not a proper attribution.

### Section 3 (Methodology)

- **(Major, Helpful Clarification)** When introducing the vocabulary $\mathcal{V}$ in Section 3.1, it would be helpful if one may preemptively indicate how such vocabulary may be generated. In this reproducibility study, this is particularly helpful since this is one of those areas where domain knowledge may be required and, therefore, the task-agnosticism claim of the original work may be challenged.
- **(Minor, Typesetting Nit)** For consistency and better typesetting, I would recommend using LaTeX’s \text{…} guard for multi-character subscripts (e.g., $\mathcal{D}_{extract}$).
- **(Minor, Typesetting Nit)** For clarity, I would suggest making external hyperlinks (such as those used in the data of CC3M) blue and underlined.
- **(Minor, Potential Typo)** “A Nvidia A100 …” should probably be “An Nvidia A100 …”
- **(Major, Clarity)** When one says something like “The qualitative support for C2 includes a human feedback survey…” the assumption is that this is something done as part of this work. However, this is in reference to the original evaluation of the proposed method by the original authors. As such, I would strongly suggest that this is made clear by avoiding the use of the passive voice in statements that can be ambiguous (e.g., this could be rewritten as “Rao et al. provide qualitative support for C2 that includes a human feedback survey…”).
- **(Minor, Accuracy)** Related to the point above, in Section 3.4.1 it is claimed that the user study by Rao et al. was replicated, however the used survey and study is different for this reproducibility study. I would recommend specifying this here as otherwise this seems contradictory to what one sees Section 3.4.3.
- **(Major, Formulation)** Shouldn’t there be a stability $\epsilon$ term in the denominator of the new term in Equation (7) as, in theory, this denominator can have zero norm due to the nature of the ReLU nonlinearity? Similarly, shouldn’t one normalize $\mathbf{v}$ in this term as well to avoid loss reductions by a simple rescaling of weights?
- **(Major, Clarity)** Could you please be more specific on how model selection is done for $C$ given the mean cosine similarity, the activation average magnitude, and the validation accuracy? In other words, given those three quantities, how was the “best” model selected (i.e., what is the cumulative metric one uses to select the best candidate)?

### Section 4 (Results)

- **(Critical, Claim)** I would disagree with the claim that because the lower-aligned concepts do not show consistent samples (Figure 4), then C2 cannot be “**reproduced**”. I would agree with a claim saying that C2 may not be as strong as a statement as put forth by the original authors, but claiming it is not reproduced implies the authors tried a similar experiment and generated very different results. This does not appear to be the case, as far as I am concerned. As such, I would recommend changing the way these (interesting) results are portrayed.
- **(Critical, Error Bars)** Against common good practices, there are no errors reported in Table 2. I would strongly recommend these error bars to be included as part of this evaluation to better understand the sensibility and variance of the method being studied.
- **(Critical, Fair Evaluation)** Why does Figure 9 show less concepts for the extended model than the reproduced? This seems rather odd and potentially unfair. I would appreciate it if it is possible to show the same number of most-relevant concepts for all models as otherwise this selection may bias the analysis.

### General Questions

- **(Major, Question)** I understand computational constraints do limit a lot of the things one can run and attempt; however, for a reproducibility study, it would be ideal to use the same setup and evaluation as used in the original paper. As such, could you please comment on why using 10% of the Places365 data does not violate the spirit of checking reproducibility on this task by following all the original steps of the work in question? Is there an alternative to using just 10% of the data?
- **(Critical, Question)** I am a bit at odds with the decision not to evaluate concept interventions. I would strongly disagree that intervention analysis only measures “human-driven interventions.” In fact, interventions are a key way of verifying quantitatively the interpretability of concepts by confirming that when a concept is modified in a certain direction, the output is updated as expected. Considering this, could the authors please elaborate on why the evaluation of interventions would not help explore and verify claim C2? As this is a core claim in both the paper and this study, I would therefore recommend that interventions, in some manner at least, are considered for this reproducibility study.
- **(Critical, Question)** Is it the case that when looking at the lower-aligned neurons in the fine-tuned model, do these make sense? Or are they still completely unaligned? In other words, does the proposed fine-tuning process also lead to better alignments in the same evaluation setup used for Figure 4?

**Strengths And Weaknesses:**

Thank you so much for submitting this work! I enjoyed reading this paper. Below are what I believe are this paper’s main strengths, followed by what I think are some of its weaknesses:

### Strengths

1. **[Significance and Originality] (Critical)** I appreciate the extensions on top of the reproducibility study. These include the extended qualitative analysis, the new alignment method, and the extended user study. All of these make this reproducibility study interesting and worth reading even if one is familiar with the original work.
2. **[Quality] (Major)** The motivation and experimental design are all very clear. More importantly, I believe the key claims examined in this paper (i.e., C1 - C3) are well-motivated and followed by a series of appropriate experiments to evaluate them.
3. **[Clarity] (Major)** The English and writing are almost typo-free, very clear, and extremely easy to read. I have some issues with the organization (as described below), but this is orthogonal to the quality of writing.

### Weaknesses

In contrast, I believe the following are some of this work’s limitations:

1. **[Quality] (Critical)** I believe some of the key claims studied in this work would’ve benefited from questioning some of the key assumptions behind the proposed methodology (e.g., the use of a fixed dictionary) and studying the reproducibility of the concept interventions results by Rao et al.. As I argue below, I do not particularly agree with the argument for not exploring the intervention results.
2. **[Clarity] (Major)** Some of the sections could’ve benefited from better subsections/splits as they seem to be conglomerates of connected but not entirely related topics. For example, see Section 3.4.2 where methodological changes as well as model selection and user study design are all discussed in the same subsection of a “Methodology” wider section.
3. **[Significance] (Minor)** This work would benefit from discussing, in a potential extended related work section, works that have come since the publication of Rao et al.. This may shed some light on how this method behaves and works, as well as on how one could potentially extend this approach. Moreover, such a section would help better place this study within the appropriate literature.

---

> ### Author Response · Authors · 2025-04-21
> **Thanks and response to concerns**
>
> Dear reviewer UByM,
>
> We would like to sincerely thank you for your time and effort in reviewing our work this thoroughly. Below, we provide a point-by-point response to your comments in the same format as the review, along with the corresponding changes made in the revised version. Please don't hesitate to reach out to us with any further questions.
>
> # Introduction
> >  when first discussing CBMs in the introduction, please cite them in the same line.
>
> We have updated the manuscript to include the appropriate citation when CBMs are first introduced in the introduction.
>
> > it would be beneficial if, when discussing “recent innovations” in the introduction, you provide valid evidence and citations of which “innovations” you are referring to.
>
> We have included two citations of papers that use GPT-3 for generating concepts and CLIP for encoding them.
>
> > (Critical, Wrong Attribution) The work of Margeloui et al. cited in the introduction does not, in any form of fashion, discuss issues of using LLMs for CBMs.
>
> We have removed the Margelou et al. citation and the faithfulness argument, instead we list general disadvantages of relying on LLMs in the context of CBMs.
>
> # Methodology
> > When introducing the vocabulary in Section 3.1, it would be helpful if one may preemptively indicate how such vocabulary may be generated.
>
> We agree with this suggestion. In the revised version, we have added the sentence: “To enable effective generalization and meaningful concept naming, the set $\mathcal{V}$ should be broad and flexible, for example, a large collection of unigrams.” The specific vocabulary used is listed in the hyperparameter table.
>
> > I would recommend using LaTeX’s \text{…} guard for multi-character subscripts (e.g., $D_{extract}$).
>
> > For clarity, I would suggest making external hyperlinks (such as those used in the data of CC3M) blue and underlined.
>
> > “A Nvidia A100 …” should probably be “An Nvidia A100 …”
>
> Thanks, fixed.
>
> > When one says something like “The qualitative support for C2 includes a human feedback survey…” the assumption is that this is something done as part of this work. However, this is in reference to the original evaluation of the proposed method by the original authors. As such, I would strongly suggest that this is made clear by avoiding the use of the passive voice in statements that can be ambiguous
>
> We agree with this suggestion and have adjusted our writing style where necessary. Example:
> *“The quantitative support for C2 includes a human feedback survey…”* —>
> *“To further investigate C2, we reproduce qualitative and quantitative analyses from Rao et al. (2024) related to interpretability. ”*
>
> > Related to the point above, in Section 3.4.1 it is claimed that the user study by Rao et al. was replicated, however the used survey and study is different for this reproducibility study. I would recommend specifying this here as otherwise this seems contradictory to what one sees Section 3.4.3.
>
> To clarify, we replicated the user study by Rao et al. exactly, with the results shown in Figure 6. Our work includes two user studies: the first mirrors the original DN-CBM paper, while the second, designed to evaluate our extension, differs in structure. The first study is described in Section 3.4.1, and the second in Section 3.4.2. After introducing the second study, we dedicate a paragraph to explaining the differences and the rationale behind them.
> Upon rereading, we understand the confusion, which stems from the ambiguity noted by the reviewer earlier. In the revised version, we will make it clearer that we conducted two distinct user studies and refer the reviewer to the revised paragraph at the end of Section 3.5 (we changed the structure to let 3.4 be about  reproducibility and 3.5 about the extension): “Note that this user study, conducted to evaluate our model extension, is the second user study in this work and differs from the first, which followed the exact structure of the study by Rao et al.
>
> Finally, we are unsure what is meant by “contradictory”, as there is no Section 3.4.3 in our original work. However, we believe the above clarifications should address this concern. Please don’t hesitate to reach out for further clarification.
>
> We will continue this comment in the next block.

---

> > ### Author Response · Authors · 2025-04-21
> > **Thanks and response to concerns (part 2)**
> >
> > >  Shouldn’t there be a stability term in the denominator of the new term in Equation (7) as, in theory, this denominator can have zero norm due to the nature of the ReLU nonlinearity?
> >
> > While the activations were never zero everywhere in practice, it is theoretically possible so thank you for pointing this out. We have included a stability term accordingly in the revised version.
> >
> > > shouldn’t one normalize $\boldsymbol{v}$ in this term as well to avoid loss reductions by a simple rescaling of weights?
> >
> > We do not normalise the vector $\boldsymbol{v}$ and want to clarify the reason:
> > We want to optimize the weighted sum of cosine scores, where each component’s cosine similarity is weighted by the corresponding activation magnitude. We normalise the activations because we don't want the model to be incentivized to arbitrarily increase the activations globally. By not normalising the cosine scores, we give the model an incentive to grow the total alignment magnitude as well as the individual directional alignments. If we were to normalise $\boldsymbol{v}$, the model would instead focus primarily on pairing high activations with relatively higher cosine scores, rather than pushing the cosine scores collectively towards 1.
> >
> > > Could you please be more specific on how model selection is done given the mean cosine similarity, the activation average magnitude, and the validation accuracy?
> >
> > We agree that the original formulation was unclear and have clarified it more explicitly in the revised version. Our paper does not include a quantitative summary statistic to determine the best value; instead, it offers general guidelines for selecting an appropriate value. For our experiments, we fixed the value of $C$ based on these guidelines to enable a focused user study, intended as a proof of concept to explore whether promoting alignment results in more faithful and interpretable model explanations.
> >
> > # Results
> > >  I would disagree with the claim that because the lower-aligned concepts do not show consistent samples (Figure 4), then C2 cannot be “reproduced” […] I would recommend changing the way these (interesting) results are portrayed.
> >
> > We agree that this may be a harsh statement. In the revised version of our work, we adopt a more nuanced approach. To clarify the original intent: In their work, Roa et al. claim explainability  C2, which is partially supported by displaying samples of figures alongside local explanations. Specifically, we believe that reproducing the quality of the local explanations shown in Figures 7, 8, D5, D6, and D7 is not feasible through random sampling of figures. The combination of these figures displays a degree of faithfulness in explanations that we were unable to match by sampling random figures.
> > That said, Rao et al. never claimed to have sampled the images for these local explanations randomly, nor do they provide a detailed discussion of their method. Furthermore, this difficulty does not discredit the proof of concept presented in their work. And more importantly, this cannot be derived from Figure 4 of our work. Therefore, we agree with the suggestion and will rephrase our claim to reflect this more carefully.
> >
> > >  there are no errors reported in Table 2. I would strongly recommend these error bars to be included as part of this evaluation
> >
> > We have added this, and now report mean and standard deviation accuracy over 3 training runs.
> >
> >
> > > Why does Figure 9 show less concepts for the extended model than the reproduced? This seems rather odd and potentially unfair. I would appreciate it if it is possible to show the same number of most-relevant concepts for all models
> >
> > We have addressed this in our paper. We added the following in the revised version:
> > Note that if two latent neurons are assigned the same label (e.g., their dictionary vectors map to the same CLIP label) and both are among the top 5 activations, they are merged into a single explanation. This is why our extended model (right) appears to provide fewer explanations. The figure still shows the top 5 activated neurons, but some share the same label. For instance, two distinct neurons labeled “horses” might both activate on the image of a horse.
> >
> > In the extended model, neurons labeled with the same concept (e.g., “horse”) are encouraged to align more closely with that concept and are pushed closer together in representation space. As a result, it becomes more likely that these neurons will activate on the same image. In contrast, while the baseline model may have the same number of neurons labeled “horse,” their representations are more varied. Some individuals may respond strongly to images of horses, while others may not, resulting in a more diverse set of top activations and fewer shared labels in explanations.
> >
> > We have one more block with comments, as we want to make sure to provide an elaborate response to all concerns and questions.

---

> ### Author Response · Authors · 2025-04-21
> **Thanks and response to concerns (part 3)**
>
> # General Questions
> > could you please comment on why using 10% of the Places365 data does not violate the spirit of checking reproducibility on this task by following all the original steps of the work in question? Is there an alternative to using just 10% of the data?
>
> Using 10% of Places365 for one downstream task in our reproducibility study does not violate the spirit of reproducibility. Our primary objective is to validate the methodology and key trends presented in the original paper, rather than to approximately match absolute performance metrics or specific figures. While this was not emphasized in the original submission, we now clarify that the 10% subset was randomly sampled and preserves the original class distribution. Moreover, the central theme of the paper,  which focuses on the interpretability of hidden neurons, is inherently independent of the specific downstream dataset used. This is exemplified by the figure, which demonstrates top-activating images for low cosine similarity concepts, even in a diverse, complete dataset such as ImageNet.
>
> > I am a bit at odds with the decision not to evaluate concept interventions. […] could the authors please elaborate on why the evaluation of interventions would not help explore and verify claim C2? As this is a core claim in both the paper and this study, I would therefore recommend that interventions, in some manner at least, are considered for this reproducibility study.
>
> We agree that concept interventions are important factors in explainability, which is why we included intervention analyses for both the reproduced and fine-tuned models. Initially, our focus was more on addressing issues related to the meaning of concepts. However, as you rightly pointed out, interventions also play a critical role in evaluating the explainability of CBMs, and we now incorporate that perspective as well.
>
> > Is it the case that when looking at the lower-aligned neurons in the fine-tuned model, do these make sense? Or are they still completely unaligned?
>
> The answer to this question is yes, though there is some nuance. We would like to comment on the general research method of task-agnosticity plots, such as the one presented in Figure 4. While we acknowledge that such visualizations can offer a preliminary indication of the success of the method, we believe they are ultimately limited in what they reveal about model explainability.
>
> This limitation is twofold:
> 1. First, the goal is to investigate the explainability of model reasoning, which we define as understanding why a model produces a particular output or classification. This objective does not align well with the methodology underlying Figure 4. Specifically, the presence of an image among the top activations for a given concept does not imply that this concept played a meaningful role in the classification of that image. Abstract or generic concepts, for instance, may consistently show a set of images, but these are not necessarily informative if the concepts themselves do not contribute to the model’s decision-making process.
>
> 2. Second, the limitations of the datasets used further compound the issue. For example, the concept “sauna” is displayed along with its top activating images, even though some datasets do not contain any actual sauna images. This creates a misleading impression of concept alignment, even when the concept itself is well-defined and aligns closely with its CLIP vector representation.
> These limitations become even more pronounced in the context of our extension, where the interpretation of “highly aligned concepts” is more nuanced. In this setting, displaying the top or bottom aligned concepts along with their corresponding top activating images, following the approach of Figure 4, becomes less meaningful. This is because many of these concepts are abstract in nature and do not substantially contribute to the model’s actual reasoning process. As a result, such visualizations may give an impression of interpretability without providing real insight into the underlying decision-making process of the model.
> Therefore, in our extension, we focus primarily on local explanations in both our figures and user study. Additionally, we reproduce the task-agnosticity plot (Figure 10) for completeness and in response to your suggestion. Additionally, we have included a paragraph discussing this, along with the nuances of the activation, in the final paragraph of Section 4.2.
>
>
> We hope to have answered all your questions and are happy to provide further elaboration if needed. Thank you again for your thorough and constructive review.

---

### Author Response · Authors · 2025-04-21
**Thanks for the reviews and key paper changes summary**

Dear reviewers,

We want to sincerely thank you for the thoughtful and constructive reviews.

As the reviewers raised a range of distinct concerns, we have provided detailed, point-by-point responses to each individual reviewer. Below, we summarize the key revisions and additions made to the paper:

* **Conclusion on C2 reproducibility revised:** We now claim C2 to be reproducible, with the note that additional analyses performed beyond the original study do not fully support C2. (In *Abstract, Results reproducing original paper* and *Discussion*)
* **Revised Introduction:** We have adjusted the introduction to account for a wrong attribution and also motivated the need for reproduction of the DN-CBM results. (In *Introduction*)
* **Improved methodology structure:** We have included a special paragraph for reproduction and a special paragraph for extensions. (In *Methodology*)
* **Extended reproduction of human intervention experiment:** We reproduced the human intervention on the Waterbirds-100 dataset by Rao et al. (2024) and extended it to the finetuned model (In *Datasets, Experimental setup and code for reproducibility, Results reproducing original paper*, *Results beyond original paper,* and *Appendix C.2*)
* **Clarified and analyzed low task-agnosticity:** We provided additional explanation of Figure 4 and examined how the finetuned SAE handles named concepts from Figure 4 (In *Results reproducing original paper*, *Results beyond original paper*)
* **Reported error terms:** We now include error values for model performance across 3 runs. (In *Results reproducing original paper*)
* **Described hyperparameter choice:** We discussed the selection of $C=10^{-4}$ for the finetuned model and some cosine similarity plots across different penalties C were moved from Appendix C.2 (Figure 18) to main text (Figure 8) (In *Results beyond original paper*)
* **Interpreted reduced number of explaining nodes:** We provided an explanation for why the finetuned SAE appears to yield fewer explaining nodes (In *Results beyond original paper*)
* **Improved flow and minor textual changes:** We have polished the text in some sections to account for the extra space needed for the other experiments. (In *Methodology*, and *Results*)

For a detailed overview of all changes made and our detailed responses to each reviewer’s comments, please refer to the point-by-point replies.

---

### Decision · Action_Editor_fgPh · 2025-05-30

**Recommendation:** Accept as is

**Comment:**

All the reviewers are satisfied with the paper and the authors' rebuttals.

**Audience:**

Researchers interested in interpretable deep learning, in particular concept bottleneck models, would be interested.

**Claims And Evidence:**

This work provides a reproducibility study of Discover-then-Name CBMs (DN-CBMs) by Rao et al. (2024). They replicate the experiments and extend the experiments to larger scale. They show that the main claims made by Rao et al. can be reproduced, but also reports an issue about faithful interpretation. They also propose a further follow-up idea that improves upon this aspect.

Reviewers raised concerns on (1) omission of human intervention experiments and (2) the potential impact of vocabulary size on performance and interpretability. The concerns were fully addressed by the authors in subsequent revisions. All the reviewers now agree that this paper provides a worthy contribution to TMLR.